

# Estimation of atmospheric total organic carbon (TOC) – paving the path towards carbon budget closure

Mingxi Yang[1*], Zoë L. Fleming[2&]

[1] Plymouth Marine Laboratory, Plymouth, United Kingdom

[2] National Centre for Atmospheric Science (NCAS), Department of Chemistry, University of Leicester, UK, United Kingdom

* Correspondence to M. Yang (miya@pml.ac.uk)

& Now at Center for Climate and Resilience Research (CR2), Departamento de Geofísica, Universidad de Chile, Santiago, Chile

**Abstract.** The atmosphere contains a rich variety of reactive organic compounds, including gaseous volatile organic carbon (VOCs), carbonaceous aerosols, and other organic compounds at varying volatility. Here we present measurements of atmospheric non-methane total organic carbon plus carbon monoxide (TOC+CO) during August-September 2016 from a coastal

city in the southwest United Kingdom. TOC+CO was substantially elevated during the day on weekdays (occasionally over 2 ppm C) as a result of local anthropogenic activity. On weekends and holidays, with a mean (standard error) of 102 (8) ppb C, TOC+CO was lower and showed much less diurnal variability. Excluding weekday daytime, TOC+CO was significantly lower when winds were coming off the Atlantic Ocean than when winds were coming off land. By subtracting the estimated CO from TOC+CO, we constrain the mean (uncertainty) TOC in marine air to be around 19 (±≥8) ppb C during this period. A proton-

transfer-reaction mass spectrometer (PTR-MS) was deployed at the same time, detecting a large range of organic compounds (oxygenated VOCs, biogenic VOCs, aromatics, dimethyl sulfide). The total speciated VOCs from the PTR-MS, denoted here as Sum(VOC), amounted to a mean (uncertainty) of 11(±≤3) ppb C in marine air. We assess the possible contributions from a number of known organic compounds present in marine air that were not detected by the PTR-MS. Future concurrent measurements of TOC, CO, and a more comprehensive range of speciated VOCs would enable a better characterization and

understanding of atmospheric organic carbon budget.

## 1 Background

The atmosphere hosts a rich variety of organic carbon, including volatile organic compounds (VOCs) such as hydrocarbons, alcohols, carbonyls, aromatics, ethers, etc, as well as lower-volatility compounds and aerosols. Some organic species contain



other functional groups, such as organosulfurs, organonitrogens, and organohalogens.  Many VOCs are reactive and affect the

atmospheric oxidative capacity, while organic aerosols are important for air quality, human respiratory health, and potentially

cloud formation.  The total number of organic carbon species in the atmosphere is estimated to be on the order of $10^4 - 10^5$

(Goldstein and Galbally, 2007).  Both the abundance and cycling of total non-methane organic carbon (here abbreviated as TOC)

are not well known.  The total reservoir of TOC in the atmosphere is recently modeled to be about 16 Tg C (Safieddine et al.,

2017), with large variability in both space and time.  Emissions of the initially reduced biogenic VOCs (including isoprene and

terpenes), estimated to be on the order of 1000 Tg yr$^{-1}$, are by far the largest terrestrial source of TOC to the atmosphere

(Guenther et al., 2012).  Anthropogenic emissions of mostly hydrocarbons and biomass burning are also important sources of

TOC (Andreae et al., 2001).  The large emissions of non-methane organic carbon relative to their relatively modest atmospheric

burden imply rapid turnover of these compounds.

Once emitted into the atmosphere, organic compounds undergo chemical reactions (gas phase and heterogeneous

oxidations, fragmentation and oligomerization, hydration and dehydration, etc) as well as physical transformations (e.g.

condensation into the aerosol phase).  Atmospheric organics are terminally removed by conversion to carbon monoxide (CO) and

carbon dioxide ($CO_2$), or wet and dry deposition to the surface as aerosols or gases (Goldstein and Galbally, 2007).  In the gas

phase, organic compounds can be photolyzed or react with oxidants such as the hydroxyl radical (OH), nitrate radical ($NO_3$),

halogen radicals, and ozone ($O_3$) at vastly different reaction rates and through distinct mechanisms.  Over periods of hours to

weeks, these compounds become progressively more oxidized in the atmosphere, yielding a wide variety of more oxygenated

compounds (Kroll et al., 2011).  Lewis et al. (2005) found that under maritime conditions at Mace Head (a coastal site in the

North Atlantic), oxygenated volatile organic compounds (OVOCs) including methanol, acetaldehyde, and acetone together

contributed up to 85% of the observed non-methane organic carbon and explained 80% of the estimated OH radical sink.  Read

et al. (2012) found that compared to the no OVOC case, the inclusion of these OVOCs led to a ~40% reduction in the modeled

OH radical concentration in the Eastern Tropical Atlantic.  Depending on levels of nitrogen oxides (e.g. NO and $NO_2$), VOCs

may be responsible for either the production or consumption of $O_3$, a harmful pollutant at high concentrations (Seinfeld and

Pandis, 2006).  Through affecting the cycling of the OH radical – the principal cleansing agent of the atmosphere and the

principal sink of the Greenhouse Gas methane, organic compounds are important for climate.

The relative importance of oxidation vs. deposition as terminal sinks of atmospheric organics is poorly quantified.

Jurado et al. (2008) estimated the global dry and wet depositions of organic aerosols to be 11 and 47 Tg C yr$^{-1}$, and wet

deposition of organic gases to be 187 Tg C yr$^{-1}$.  The ocean is found to be a source of organic gases such as dimethyl sulfide

(DMS, e.g. Lana et al. 2011), OVOCs (e.g. Yang et al. 2014; Schlundt et al. 2017), isoprene (e.g. Hackenberg et al. 2017),

methane ($CH_4$, e.g. Forster et al. 2009), and halocarbons (Yokouchi et al., 2013), with annual fluxes of tens of Tg C yr$^{-1}$ or less.



Thus far, the few estimates on atmosphere-ocean diffusive total organic gas transport (Dachs et al. 2005; Ruiz-Halpern et al.

2010; Hauser et al. 2013; Ruiz-Halpern et al. 2014) have yielded net deposition and emission fluxes that are more than an order

of magnitude greater than the fluxes above, illustrating the substantial uncertainty in the role of the ocean on the atmospheric

TOC budget.

        Simultaneous quantification of all atmospheric organic species individually is essentially impossible (or not always

desirable), hence the need to measure total organic carbon as a single parameter (Heald et al. 2008).  Several commercially

available methods have been developed to measure total gaseous non-methane hydrocarbon concentrations in relatively polluted

environments.  However, these methods have large uncertainties and/or poor precision.  Direct gas chromatography/flame

ionization detection (GC/FID), photoacoustic infrared (PA-IR), and photoionization detector (PID) techniques all suffer from

variable instrumental response towards different organic compounds.  Thus, without prior knowledge of the atmospheric

composition, accurate quantification of the total organic carbon with these methods is impossible.

        To achieve equal instrument response regardless of the compound type, organic carbon may be oxidized and detected as

$CO_2$ (analogous to measurement of dissolved organic carbon in water).  Roberts et al. (1998) developed a technique based on

cryogenic trapping of TOC, pre-separation of the major background gases ($CO_2$, $CH_4$, and CO) from TOC by a GC column,

thermal desorption and catalytic conversion of TOC first to $CO_2$ and then to $CH_4$, and finally detection by flame ionization

detection.  As cryogenic trapping retains moisture, sensitivity towards changing ambient humidity and variability in pre-

separation/desorption contribute to uncertainties in their system (on the order of 10 ppb C).  A similar method was developed by

the group of Paulson et al. 2002, Maris et al. (2003), and Chung et al. (2003), with a detection limit of 35 ppb C.

        Efforts trying to close the atmospheric organic carbon budget have often yielded significant fractions of 'missing', or

unidentified carbon.  Roberts et al. (1998) reported TOC concentrations from a remote site in Nova Scotia, Canada that were

typically ~30% higher than the sum of the speciated compounds.  In air quality studies in California, Chung et al. (2003) found

that measurements of total non-methane organic carbon and speciated VOCs by standard GC/FID agreed well near primary

pollution sources.  However, in aged air masses the total non-methane organic carbon was in excess by up to 45%.  Similarly,

observations of total OH reactivity when evaluated with speciated organic concentrations often suggest that a large fraction of

OH reactivity is unexplained, e.g. a portion of the total organic carbon is not detected by speciated measurements (Nolscher et

al., 2016).

        Hunter et al. (2017) attempted to close the atmospheric organic carbon budget at a forested site by simultaneously

deploying five mass spectrometers, a Herculean undertaking.  They found that previously unmeasured species such as

semivoltatiles and intermediate-volatility organics accounted for a third of the sum of observed organic carbon.  Isaacman-

VanWertz et al. (2018) used the same set of techniques to track the oxidation chemistry of a monoterpene over multiple



generations in the laboratory. They found that after a day of atmospheric ageing, most of the organic carbon ends up as either

VOCs or organic aerosols. Safieddine et al. (2017) modeled the global atmospheric TOC budget, which is found to be

dominated by ketones, alkanes, alkenes, and aromatics. Both comprehensive speciated measurements as well as modeling

studies of atmospheric organic carbon budget can benefit from in situ measurements of TOC, which are to date extremely scarce.

In this paper, we describe a simple and robust method of quantifying TOC, and show how TOC observations from a coastal site

varied with concurrent speciated VOC measurements and other environmental variables.

## 2 Experimental

Observations of total and speciated organic carbon were made from the rooftop of Plymouth Marine Laboratory (PML, ~45 m

above mean sea level, ~300 m from waters edge) during August-September 2016. Similar to the setup used by Yang et al.

(2013), ambient, unfiltered air was pulled from an inlet on the rooftop via 25 m of unheated perfluoroalkoxy (Teflon PFA) tubing

(6.4 mm inner diameter, ID) at a flow rate of ~30 L min$^{-1}$. TOC system and the proton-transfer-reaction mass spectrometer

(PTR-MS) subsampled from this main flow at a flow rate of ca. 140 and 120 mL min$^{-1}$, respectively. The TOC system was

operated continuously during this five-week campaign. The PTR-MS measurements were made primarily after working hours

and on weekends, as the instrument was needed by other projects during the working hours.


### 2.1 The TOC System

TOC was quantified in a simple fashion – via catalytic oxidation of organics in bulk air to $CO_2$. Ambient air was directed

through a platinum catalyst periodically (typically 2 minutes of ambient air followed by 1 minute of catalyst air) via a 3-way

polyterafluoroethylene (PTFE) solenoid valve (Takasago Electric, Inc.). Platinum on glass wool (Shimazu) was packed in a 13

mm diameter stainless steel tube and heated to 450°C. Dry $CO_2$ and $CH_4$ mixing ratios were continuously monitored at a

frequency of ~2 Hz by a Picarro G2311f $CO_2$/$CH_4$/H2O analyzer (in high precision mode). The Picarro instrument was

calibrated against NOAA $CO_2$ and $CH_4$ standards.

From each pair of ambient and catalyst measurements, the sum of TOC and CO is derived semi-continuously in mixing

ratio of carbon:

$$TOC + CO = CO_2^* + CH_4^* - CO_2 - CH_4 \tag{1}$$

Here $CO_2$ and $CH_4$ represent the dry mixing ratios in ambient air (taken from the last 5 seconds before switching to measuring

catalyst air). $CO_2^*$ and $CH_4^*$ indicate dry mixing ratios when air is directed through the catalyst (taken from the last 25 seconds

of the catalyst stage). The monitoring of $CH_4$ enables the continuous assessment of the efficiency of the catalytic conversion

(computed as $[CH_4 - CH_4^*]/CH_4$). At the flow and temperature used, oxidation of $CH_4$ was highly efficient (98.7-98.9%) and




l20    largely insensitive towards humidity during this campaign (Figure 1). $CH_4$ is thermodynamically one of the most reduced and

stable compounds. Thus its rapid and near complete removal by the catalyst suggests ~100% oxidation of other VOCs as well as

CO to $CO_2$, which is confirmed by laboratory tests as discussed below. In addition to gases, ambient TOC measured with this

method likely includes some aerosols and low/moderate-volatility compounds. The contribution of particulate and semi-volatile

organics towards TOC depends on their transmission through the inlet tube as well as on their oxidation efficiency in the

l25    catalyst. We did not test these aspects as organic aerosol mass is already quantifiable using aerosol mass spectrometry as well as

thermal methods (e.g. Sunset Laboratory's OCEC analyzer), and thus not the focus of this work.

   To verify the TOC system, we measured a diluted VOC gas mix. A multi-species gas standard consisting of methanol,

acetaldehyde, acetone, DMS, benzene, and toluene (nominal mixing ratio of 500 ppb for each VOC balanced in nitrogen, Apel-

Riemer Environmental, Inc, USA) was diluted by a factor of 10 with zero air. The zero air was generated by pre-scrubbing a

l30    low-VOC synthetic air (BOC BTCA 178, containing 20% oxygen) with a second 450°C platinum catalyst.

   The expected total ppb of carbon in this diluted standard (1295 ± ≤78) ppb is computed as follows:

$$Sum(VOC) = \sum VOC \cdot N_c$$

  (2)

Here $N_c$ is the number of carbon in each speciated VOC. The total uncertainty in this Sum(VOC) is propagated from the

accuracies of the VOC standard concentrations and the uncertainties in the dilution. We measured a difference in TOC + CO

l35    between the diluted VOC standard and zero air alone of 1232 (± 1 standard error of 21) ppb. Assuming negligible CO in the

zero air as well as in the VOC standard, TOC + CO here can simply be equated to TOC. We see that TOC and Sum(VOC) in

this test agree well within the experimental uncertainties.

   Because the catalyst (made up of platinum, glass wool, and stainless steel) does not contain any carbonaceous

components, we expected the instrument background in TOC + CO (i.e. when measuring air that is free of organics and CO) to

l40    be zero. However, post-campaign measurements of zero air (see above) yielded a TOC + CO background of 39 (± 1 standard

error of 3) ppb. The reason for this small but significantly positive background is unclear. It could be that some particulate

organic carbon either preexisting in the atmosphere or formed via charring to 450°C is captured by the glass wool, and then

slowly oxidized to $CO_2$ over time. We note that the Sunset Laboratory's OCEC (Organic Carbon Elemental Carbon) analyzer

heats to 850°C (over manganese dioxide) for complete desorption and conversion of refractory organics (e.g. soot) to $CO_2$. We

l45    subtracted the background value of 39 ppb from the TOC + CO measurements during the 2016 campaign. However, the fact that

the background measurement was not made at the time of the campaign is a source of potential bias in this dataset.

   Measurement of TOC+CO by our approach is made possible thanks to the very high precision of the Picarro G2311f

instrument (~100 and 0.4 ppb for $CO_2$ and $CH_4$ at 2 Hz, respectively). Scatter in TOC+CO depends on random noise as well as

ambient variability in the $CO_2/CH_4/CO$ mixing ratios. By averaging, random noise can be reduced and the measurement





precision significantly improved. In the limit of no ambient variability in $CO_2/CH_4/CO$, each pair of $CO_2*/ CH_4*$ and $CO_2 /CH_4$

measurements has a propagated uncertainty of about 35 ppb in our setup. This implies a best-case hourly precision of 8 ppb C

for TOC + CO – a value that may be appropriate for parts of the remote marine atmosphere (i.e. little variability in

$CO_2/CH_4/CO$). Closer to emission sources, the greater variability in $CO_2/CH_4/CO$ (especially $CO_2$) is expected to significantly

worsen the precision in TOC + CO. At our polluted coastal environment, the standard deviation in the hourly mean TOC + CO

was about 30 ppb during hours of fairly low ambient $CO_2$ variability (1 standard deviation of ~0.2 ppm). While not significantly

more precise than earlier methods, the technique described here is robust in that it avoids many of the uncertainties and

complexities associated with trapping and desorption (e.g. Roberts et al. 1998).

**2.2 Speciated VOC measurements**

Speciated organic gases were quantified using a PTR-MS, which was freshly serviced and calibrated by Ionicon. The PTR-MS

settings were essentially the same as those used by Yang et al. (2013; 2014), except for a lower drift tube pressure (2.25 mbars).

The monitored masses (m/z) with a $H_3O^+$ source were attributed to the following compounds: m/z 33 (methanol), 42

(acetonitrile), 43 (fragmented acetic acid or propanol), 45 (acetaldehyde), 47 (ethanol), 59 (acetone), 61 (acetic acid or propanol),

63 (DMS), 69 (isoprene), 79 (benzene), 91 (toluene), 107 (xylene), 137 (monoterpenes). The total speciated VOCs from the

PTR-MS is computed following Equation 2. The same VOC gas standard as above (containing methanol, acetaldehyde, acetone,

DMS, benzene, and toluene) diluted in synthetic air (BOC BTCA 178) was used for 1) calibration of the PTR-MS, and 2) further

demonstration of the oxidation efficiency of the platinum catalyst. For detected VOCs that were not directly calibrated,

recommended kinetic reaction rates from Zhao and Zhang (2004) were used to compute the mixing ratios from the PTR-MS raw

counts.

The oxidation of all VOCs in the platinum catalyst tested here appears to be immediate, complete, and independent of

the amount of VOCs present as well as the humidity (within the test range). Figure 2 shows a time series of raw VOC mixing

ratios from such a laboratory experiment. The flow of the VOC gas standard was increased stage-wise, resulting in greater

measured VOCs from about 3000 to 11500 seconds. The differences in the amplitudes of the different raw VOC mixing ratios

are primarily due to their different kinetic reaction rates and transmission efficiencies within the PTR-MS, which are corrected

for in the final dataset. For the second half of each stage of dilution, the VOC containing air was directed through the platinum

catalyst. With increasing amount of VOCs added, VOC measurements during the catalyst section remained constant and equal

to (or slightly less than) measurements of synthetic air alone (after ~12200 seconds), demonstrating 100% oxidation efficiency.

In a separate experiment, synthetic air was moistened to an absolute humidity of 12.8 g kg$^{-1}$ before the addition of the VOC

standard. The higher humidity in the sample air did not affect the oxidation efficiency of these VOCs.





Within the range of ambient humidity during this campaign and under the PTR-MS setting used, the only VOC

measurements with noticeable humidity-sensitivity were isoprene and monoterpenes. Calibrations and air scans show that these

two VOCs were partially fragmented into m/z of 41 and 81, respectively, consistent with previous findings (e.g. Schwarz et al.

2009; Tani et al. 2004). The fragmentation tended to be more severe at a lower ambient humidity and we account for these

daughter fragments in the budgets of biogenic VOCs and Sum(VOC). We show in Section 5 that Sum(VOC) was dominated by

OVOCs during this campaign and these gases were well calibrated. Biogenic VOCs accounted for only a small fraction of

Sum(VOC), such that any uncertainty from the humidity-dependence in these fragmentations likely contributed little to

Sum(VOC). In total we expect the accuracy in Sum(VOC) during our campaign to be at worst 25%.

**3 Variability in TOC+CO**

The period of 4$^{th}$ August to 22$^{nd}$ September 2017 was mostly sunny (noontime shortwave irradiance of 600-700 W m$^{-2}$) and calm

(mean wind speed ~4 m s$^{-1}$). Figure 3A shows the time series of hourly averaged TOC+CO from the entire measurement period.

Large variability was observed in TOC+CO on weekdays, with values occasionally exceeding 2 ppm C. In contrast, TOC+CO

was significantly lower and less variable on weekends and holidays. Stronger winds up to 13 m s$^{-1}$ from the southwest came

through between 19$^{th}$ and 22$^{nd}$ August and on 3$^{rd}$ September, carrying along considerable rainfall (Figure 3B). Relative low and

consistent TOC+CO were observed during these periods.

The hourly TOC+CO data were clearly noisier in the first week, when the measurement frequency was lower (5 minutes

of ambient air followed by 2 minutes of catalyst air). Increasing the measurement frequency subsequently (2 minutes of ambient

air followed by 1 minute of catalyst air) reduced the noise in the hourly averaged TOC+CO by about 50%. Scatter in the hourly

TOC+CO (and some of the negative values), largely a result of highly variable ambient $CO_2$ mixing ratios, is greatly reduced in

the 6-hour mean and median. This further confirms that our TOC + CO measurement is precision-limited.

Averaged over the entire campaign, TOC+CO was much higher (exceeding 1 ppm C) during the day than at night

(~0.15 ppm C) on weekdays (Figure 4A). Typical of urban air pollutants, two peaks were observed in the weekday data, one

around the morning rush hour traffic, and the second one around the late afternoon rush hour traffic. Less diurnal variability was

apparent on weekends and holidays, with mean (standard error) TOC+CO of 102 (8) ppb C. Higher ambient $CO_2$ mixing ratio

was also generally observed on weekdays than on weekends and holidays during the daytime (Figure 4B). This suggests that the

elevated TOC+CO levels during the weekday daytime were largely due to local anthropogenic activities. In particular,

demolition of a large building with heavy machinery took place south of the PML building during the working hours of this

period.



We estimate the emission factor of TOC+CO from local anthropogenic sources. The difference in TOC+CO between

weekdays and weekends/holidays is plotted against that of $CO_2$ in Figure 5. A positive correlation is evident, with a

dimensionless ratio of roughly 0.1 (largely between 0.05 and 0.2). This is on the higher end of previously reported $CO:CO_2$

emission ratios from urban environments (e.g. Wang et al., 2010; Ammoura et al., 2014), probably in part due to the inclusion of

VOCs.

For the rest of this paper, we focus on data from weekends/holidays and weekday nighttime only in order to minimize

the effects of local pollution. TOC+CO and $CO_2$ are averaged in wind direction bins in Figure 6A. Both variables show

qualitatively similar patterns, with lower values when winds were from the sea (southwest, here 180-270°) than when winds were

from land (north to southeast, here 330-120°). Comparing the difference between land air and marine air in TOC+CO (0.247 –

0.079 ppm C) vs. that in $CO_2$ (421.0 – 396.4 ppm), we get a dimensionless ratio of approximately 0.0068. This ratio is about an

order of magnitude lower than the slope derived from local pollution (Figure 5) for probably multiple reasons. Unlike local

pollution, TOC+CO in marine air is likely less coupled to $CO_2$ emissions. Due to the shorter lifetimes of organics, we also

expect a far more rapid degradation of TOC further away from the emission sources than $CO_2$.

A fraction of TOC measured at PML appeared to be highly soluble and readily removed by wet deposition. The mean

(SE) TOC+CO during non-raining periods was 0.136 (0.008) ppm C, while that during rain events (65 hours total over the

campaign) was 0.071 (0.002) ppb C. Though part of this difference could be due to the greater occurrence of rain when winds

were from the southwest. TOC+CO correlated in the mean with $CO_2$ (Figure 7A) and also with $O_3$ (Figure 7B) from the Penlee

Point Atmospheric Observatory (PPAO) 6 km to the south of PML (Yang et al., 2016), similar to observations from Roberts et

al. (1998). For reference, mean $CO_2$ and $O_3$ were about 395 ppm and 18 ppb when winds came from the southwest during this

campaign. In the following sections, we constrain the magnitude of TOC in marine air and compare it against speciated organic

carbon measurements from the PTR-MS as well as against previous observations.


**4 Constraining the magnitude of TOC**

Accurately determining TOC requires the concurrent measurement of CO, which was unfortunately not available during this

period. CO was measured at PML during May-July 2017 using a combination of gas chromatography and chemical sensors by

the University of York. Averaged to daily bins, these measurements agreed within ~10% in the mean with CO measured from

the Defra Air Quality monitoring station in Cardiff during the same period when winds were from the southwest (Katie Read,

personal communications in 2018). Furthermore, the Copernicus Atmospheric Monitoring Service (CAMS) analysis of CO

(Inness et al., 2015), constrained by satellite column as well as in situ CO measurements, generally do not show a large



difference in surface CO mixing ratio between Plymouth and Cardiff. This is not surprising because these two maritime cities

both face the Atlantic to the southwest and are only about 150 km part.

We subtract CO measured at Cardiff during August-September 2016 from TOC+CO to yield TOC. Doing so is likely

only reasonable during southwesterly conditions. TOC + CO from the southwest sector in our measurements had a mean (SE) of

79 (7) ppb, while CO at Cardiff during these conditions averaged 60.3 (SE of 1.5) ppb. This implies in a mean TOC mixing ratio

of about 19 ppb C in marine air (with a minimum uncertainty of 8 ppb). Our estimate is within the range of previous

observations. Roberts et al. (1998) reported a median TOC mixing ratio of 55 ppb C from the city of Boulder, Colorado and 11

ppb C from the remote Chebogue Point, Nova Scotia. Chung et al. (2003) measured TOC mixing ratios of several hundred ppb

C from polluted areas of California.

          To further minimize any local land influence, we focus on airmasses coming off the Atlantic Ocean. The UK Met

Office's NAME dispersion model (Jones et al., 2007) was used to produce a footprint of the air arriving at the station every 3

hours. The residence time over the 5-day journey of airmasses over a series of geographical regions were calculated, and then

converted into % relative residence times out of the whole domain. As shown in Figure 7C, TOC+CO decreased with increasing

Atlantic influence. In Atlantic-dominated (>80% relative residence time) airmasses, TOC + CO had a mean (median) of 81 (65)

ppb C, in agreement with the statistics selected by the southwest wind sector. This again implies that TOC in Atlantic-dominated

airmasses averaged about 20 ppb C. In contrast, in airmasses dominated by mainland Europe and the English Channel (>50%

relative residence time) TOC+CO had a mean (median) of 198 (191) ppb C (Figure 7D). Examples of these two types of

airmasses as well as the regional map used for the airmass classification are shown in Figures S1-S3.

### 5 Attempting a TOC budget closure

During the 1.5 months study, the PTR-MS was used to measure a large range of organic gases over 22 days. The total mixing

ratio of speciated organic carbon, Sum(VOC), is shown in Figure 8, which averaged 15 ppb C. Sum(VOC) was higher during

weekdays (mean ± standard error of 16.0 ± 0.4 ppb C) than on weekends (mean ± standard error of 14.3 ±0.4 ppb C), but this

difference is much less drastic than in the case of TOC+CO. This suggests that the significantly elevated TOC + CO during the

weekday daytime was largely due to compounds that were not measured by the PTR-MS (e.g. CO, alkanes, or alkenes).

          When limiting data to weekends and weekday nights only, Sum(VOC) correlated positively with TOC+CO in the mean

(Figure 9). Sum(VOC) was dominated by OVOCs (here methanol, acetone, acetaldehyde, ethanol, acid acid/propanol),

consistent with Lewis et al. (2005) and Heald et al. (2008) (Figure 6B). Aromatic compounds (benzene, toluene, xylenes) were

more abundant when winds were from land (northwest to northeast), as expected from anthropogenic emissions. Similarly,

mixing ratios of biogenic VOCs (isoprene monoterpenes) were higher when winds were from northwest to northeast as



compared to south/southwest. Among detected compounds, only DMS mixing ratio was higher in marine air than in continental air. Similar to TOC+CO, Sum(VOC) showed the lowest value when winds were from the sea (~11 ppb C during southwesterly conditions). In comparison, in the synthesis by Heald et al. (2008) the sum of speciated organic compounds was about 8, 14, and 18 ppb C at Trinidad Head (California), Azores, and Chebogue Point (2004 measurements), respectively. Our mean Sum(VOC) is within the range of those coastal observations, which were generally more comprehensive than just the PTR-MS measurements here. Interestingly, the approximate carbon fraction of total VOCs (i.e. carbon mass : total mass) was also the lowest when winds were from the sea. This suggests that VOCs in marine air on average contain more non-carbon functional groups (e.g. nitrogen, sulfur).

We see that TOC (19±≥8 ppb C, estimated by subtracting CO from TOC+CO) is ~70% higher than Sum(VOC) when winds were from the southwest but the difference is within the measurement uncertainties. Comparing TOC with Sum(VOC) in background marine air, necessitated here by our lack of in situ CO observations, challenges the signal to noise of the TOC measurement. We expect such a comparison to be more insightful in environments with a higher TOC burden. Nevertheless, we know that the PTR-MS with hydronium ion source is not suitable for detecting many VOCs, such as low molecular weight hydrocarbons that are expected to make up most of the primary anthropogenic organic emissions. Below we examine the magnitudes of some nominally abundant VOCs in marine air that were not measured by the PTR-MS with a hydronium source.

Formaldehyde (HCHO), the most abundant aldehyde in the atmosphere, was measured at the PPAO from the spring of 2015 to the beginning of 2016 using MAX-DOAS. During southwesterly conditions the surface mixing ratio of HCHO was about 0.5 ppb C (Johannes Lampel, personal communication in 2016). Non-methane hydrocarbons, such as alkanes, have large seasonal variability in temperate regions, with significantly lower abundance in the summer time due to greater OH destruction. Grant et al. (2011) reported long-term time series measurements of hydrocarbons from Mace Head. For the months of August and September from 2005 to 2009, the mean mixing ratio of ethane and propane were about 2 and <0.3 ppb C during maritime conditions, respectively. Grant et al. (2011) also measured (i- and n-) butane and (i- and n-) pentane, which were of even lower abundance. Salisbury et al. (2001) reported the mixing ratios of a large range of alkenes from Mace Head. The most abundant alkenes, ethene and propene, had mean mixing ratios of about 0.05 and 0.06 ppb C, respectively. Acetylene, the simplest alkyne, has a mixing ratio of ~0.5 ppb C over the Atlantic (Xiao et al., 2007). The ocean is a source of a host of halocarbons. Among these chloromethane ($CH_3Cl$) is the most abundant, with a mixing ratio of the order of 0.5 ppb C in the marine boundary layer (Yokouchi et al., 2013). Together these organic gases make up to approximately 5 ppb C.

Because of the high temperature and oxidative conditions in the catalyst, we expect some organic aerosols and semi-volatile species to be oxidized and detected as $CO_2$. Particulate matter less than 2.5 µm (PM2.5) in the City Center of Plymouth, also from the Defra Air Quality Monitoring station, was about 6 µg m$^{-3}$ during this period when winds were from the southwest.



If we assume that half of the PM2.5 was made up of carbon by mass (most likely an overestimate for this region, e.g. Morgan et al., 2010), the aerosol contribution to TOC could be up to 6 ppb C (or a third of TOC). Heald et al. (2008) reported that organic

aerosols only accounted for 4-16% of total speciated organics (gases and aerosols) in the marine atmosphere. Overall, considering the aforementioned VOCs and organic aerosols that were not detected by the PTR-MS, there does not appear to be a substantial 'missing' term in the TOC mass budget.

**6 concluding remarks**

In this paper we report a relatively novel and simple method to measure the mixing ratios of total organic carbon (TOC) and carbon monoxide in the atmosphere at a high frequency. Based on essentially complete oxidation of organics in bulk air to $CO_2$, the method shows very low sensitivity towards ambient humidity, avoids the complexities associated with trapping and desorption, and has an hourly precision of as low as 8 ppb C. Future measurements with and without an aerosol filter and a heated inlet should enable the semi-volatile and particulate fractions of TOC to be separated from the VOCs.

The estimated TOC from a polluted marine environment is compared to the sum of speciated VOCs here. Accounting for literature values of unmeasured VOCs and organic aerosols, there does not appear to be a significant undetected fraction of organics in marine air. A more rigorous examination of the atmospheric organic carbon closure requires concurrent measurements of TOC, CO, and a comprehensive range of speciated organic compounds. Additional measurements of total OH reactivity would bridge the gap between organic burden and composition with oxidative capacity. With recent advances in mass

spectrometry that are able to resolve ever more organic species (Hunter et al., 2017) as well as in chemical transport modeling (Safieddine et al., 2017), the stage seems set for closing the atmospheric organic budget.

**Acknowledgment**

This work is a contribution to the NERC project ACSIS (The North Atlantic Climate System Integrated Study). We thank K.

Read (University of York) for CO measurements at PML; M. Panagi (University of Leicester) for creating the regional map used for airmass classification; C. Wohl and D. Phillips (PML) for calibration of the PTR-MS; F. Hopkins, T. Bell, P. Nightingale, and T. Smyth (PML) for operational support; A. Rees, and I. Brown (PML) for the use of NOAA gas standards; M. Yelland and R. Pascal (National Oceanography Centre) for letting us use the Picarro instrument; T. Bertram (U. Wisconsin) for helpful discussions. Finally, we thank anonymous reviewers' comments on the first iteration of this manuscript. © Crown 2018

copyright Defra via uk-air.defra.gov.uk, licensed under the Open Government Licence (OGL). Thanks to the Met Office for use of the NAME model and the STFC JASMIN computer for hosting the model.





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

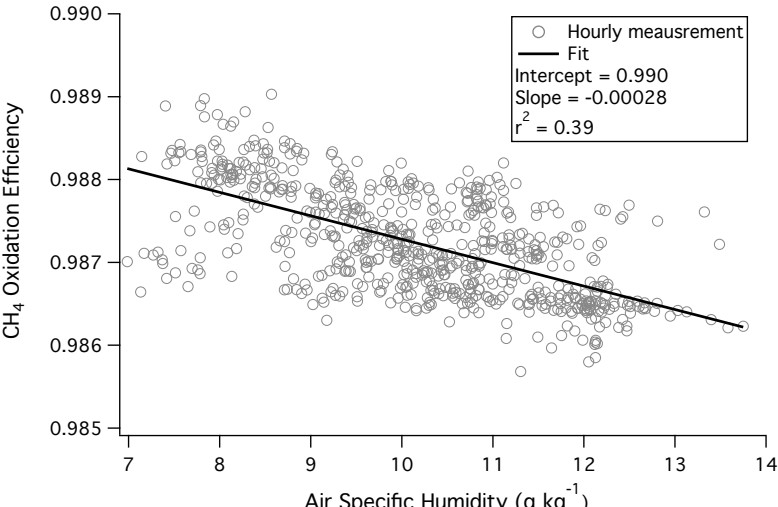

Figure 1. Oxidation efficiency of CH$_4$ by the platinum catalyst was nearly 99% and demonstrated only a weak dependence on ambient humidity.

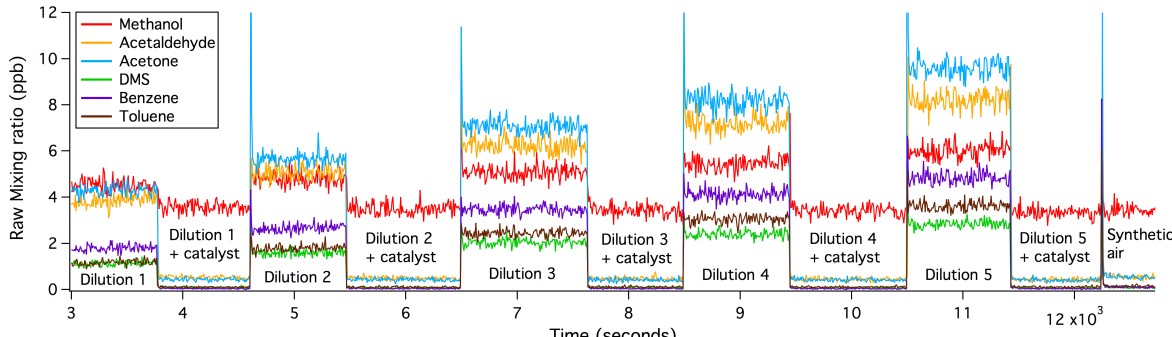

Figure 2. Catalytic oxidations of VOCs were complete, immediate, and independent of the VOC input within the range tested.






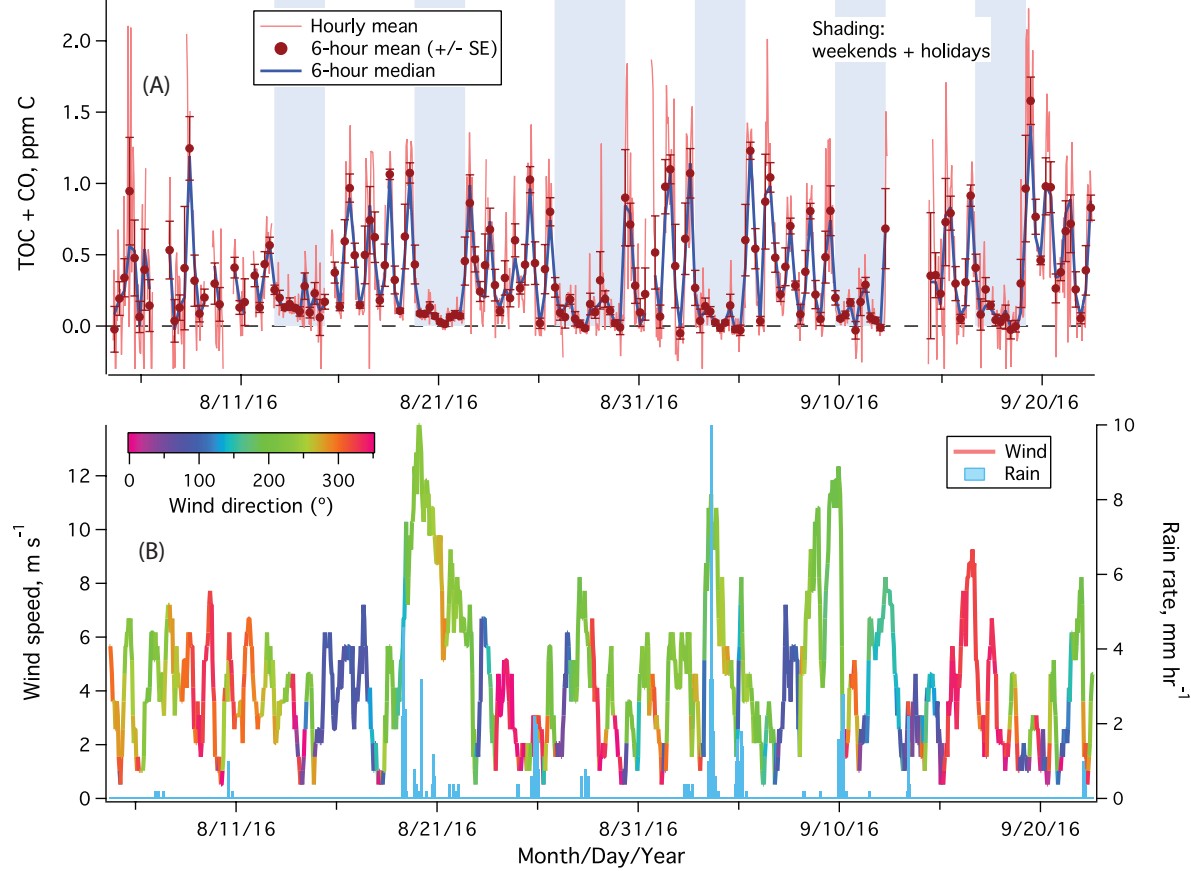

Figure 3. (A) Time series of TOC+CO in units of ppm C. Error bars on 6-hr means indicate standard errors; (B) wind speed (color-coded by wind direction) and rain rate.






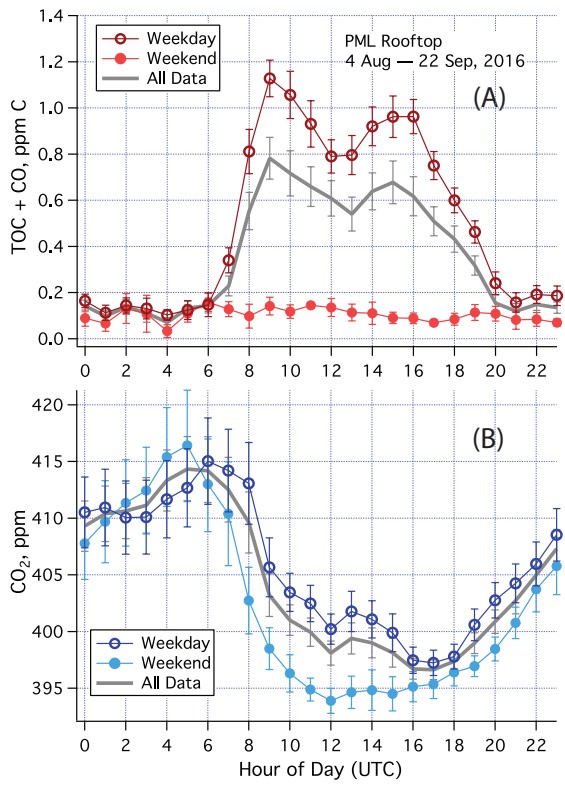

Figure 4. (A) Diel cycle in TOC+CO, and (B) $CO_2$ for weekdays, weekends/holidays, and all data. Error bars indicate standard errors. TOC+CO were much higher during weekdays than weekends, especially in the daytime. Limited diel variability in TOC+CO was observed in the weekend data. $CO_2$ was also higher in the daytime during weekdays than weekends. UTC here is one hour behind local time.

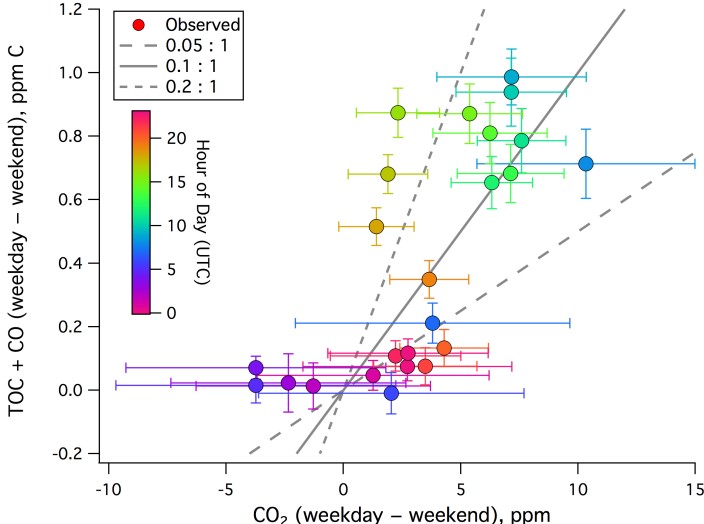

Figure 5. Weekday-weekend difference in TOC+CO (ppm C) vs. weekday-weekend difference in $CO_2$. A positive correlation is observed, with a dimensionless ratio ranging from mostly less than 0.05:1 to over 0.2:1.





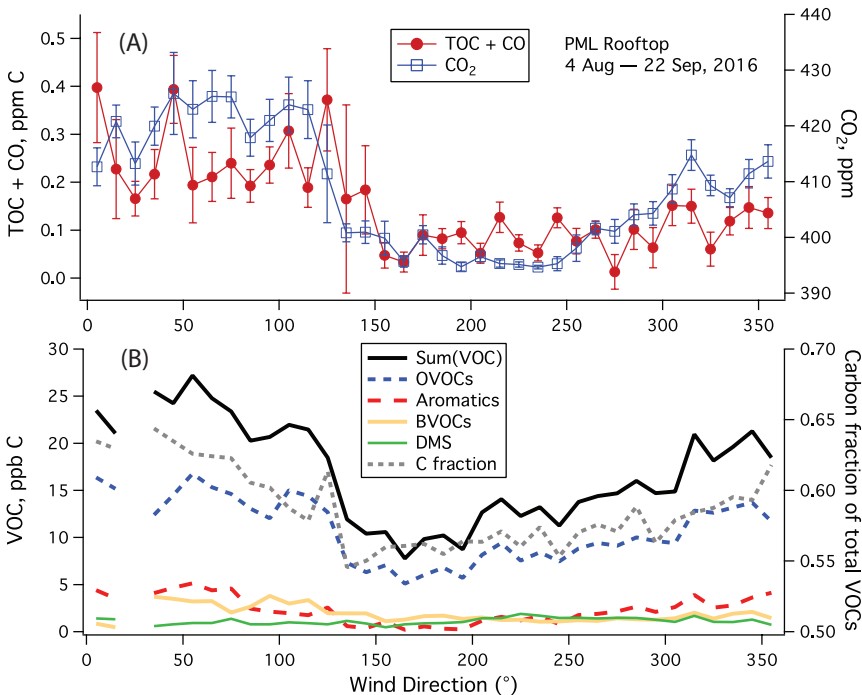

Figure 6. (A) TOC+CO and $CO_2$, and (B) Sum(VOC) and speciated VOC averaged to 10-deg wind direction bins. Error bars indicate standard errors. TOC+CO, $CO_2$, and Sum(VOC) showed higher values when winds were from land (north to southeast) than winds were from the ocean (southwest). Most VOCs had lower mixing ratio in marine air than in air from land except for DMS. Amongst speciated VOCs measured by the PTR-MS, OVOCs dominated in terms of carbon mass. The fraction of carbon in the total speciated VOC mass was also lower in marine air.





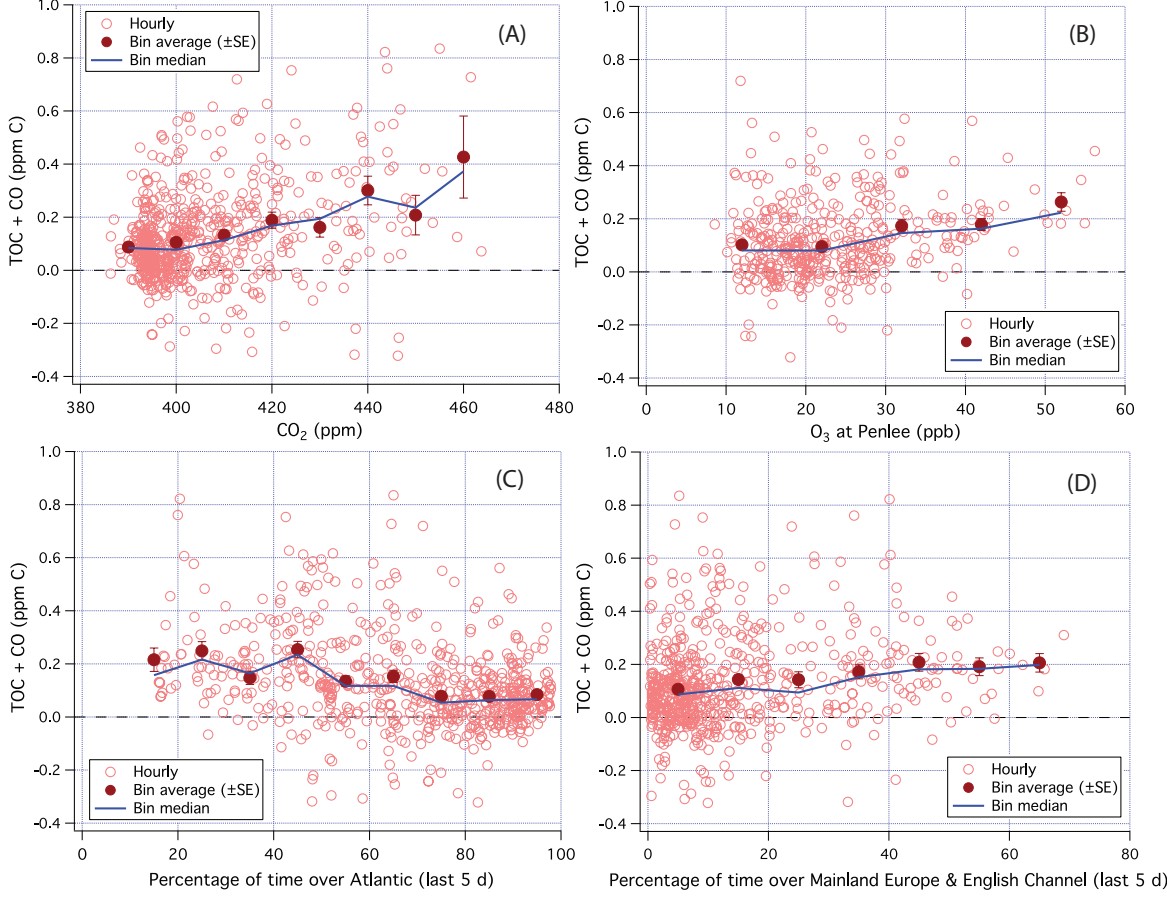

Figure 7. Relationship between TOC+CO with (A) $CO_2$; (B) $O_3$; (C) percentage of time that the airmass was over the Atlantic ocean over the last 5 days; and (D) percentage of time that the airmass was over Mainland Europe and the English Channel over the last 5 days. Hourly data limited to weekends & weekday nights only.



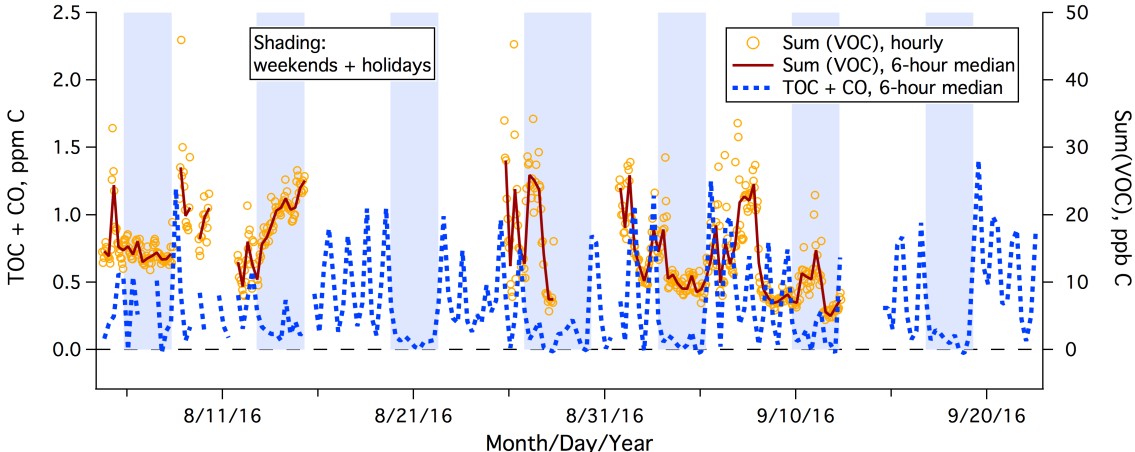

Figure 8. Time series of TOC + CO and Sum(VOC). Sum(VOC) is shown as hourly mean and 6-hour median, while for clarity TOC + CO is shown as 6-hour median only.

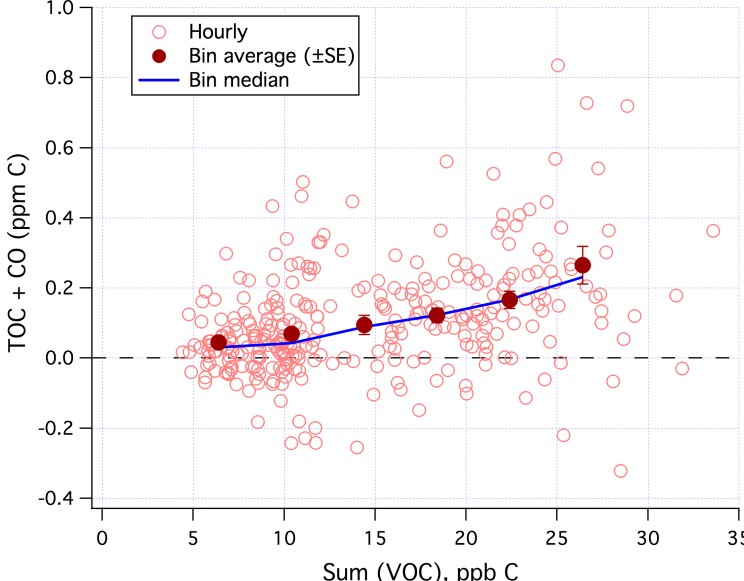

Figure 9. TOC+CO correlated positively with Sum(VOC). Hourly data limited to weekends & weekday nights only.
