# Peer review of "Estimation of atmospheric total organic carbon (TOC) – paving the"

_Atmospheric Chemistry and Physics, 2018_

## Referee Comment (RC1) · Anonymous Referee #1 · 5 Nov 2018

The authors employ a new catalyst based approach to measuring TOC+CO in ambient air (with co-location of high-precision measurements of CO2 and CH4 which allow the subtraction of these species). Measurements of TOC are much-needed and though there are limitations to these specific measurements, this manuscript describes an interesting new approach to TOC measurements and a straight-forward application to a marine site.

I have only minor suggestions and requests for clarification:

1. Given the novel measurement approach it seems this should be described in the abstract (brief description, precision, comment on whether all species are comprehensively detected – see comment #2).

2. Section 2: The authors mention the inlet and briefly allude to SV/IVOCs and aerosols in the text (lines 100, 123-126). The manuscript would benefit from more discussion of this, but most importantly, the authors should reiterate these gaps in the conclusions and abstract. Ultimately the reported TOC is not comprehensive and this should be made clear to the reader, with appropriate suggestions for assessing the degree of comprehensiveness and/or improving the instrumentation in the future (as given on lines 308-309).

3. Section 2.1: There is some ambiguity of units in this section between ppb and ppbC and it would be useful for the units of all quantities to be carefully defined (line 115). I believe that some quantities are incorrectly given as ppb instead of ppbC (line 131, 135, 141, 145, 148, 151, 155), though it's not always clear from the text. Please correct as necessary.

4. line 84: I suggest you place "e.g" in front of the Nolscher et al. reference since many studies have discussed the "missing OH reactivity"

5. line 194: September 8-10 also looks windy and rainy from the plots. Why aren't these dates included here?

6. Figure captions: I recommend adding the measurement location to the captions so that the casual reader is clear that these represent field measurements at a given site.

7. Figure 7, Figure 9, and lines 225-226, 250, 263: These scatter plots show some relationships, but the correlation appears quite weak. Please include the $R^2$ on the figures and temper the text accordingly.

---

## Referee Comment (RC2) · Anonymous Referee #2 · 13 Nov 2018

**Review of: Estimation of atmospheric total organic carbon (TOC) – paving the path towards carbon budget closure**
**Yang and Fleming, 2018**
**acp-2018-1055**

**Summary:**
The authors measured total organic carbon using a Picarro CH4/CO2 analyzer in combination with a platinum catalyst in an urban area in the UK for one month. The total organic carbon was compared to speciated VOC from PTR-MS. A weekday/weekend effect was discovered. The sum of speciated PTR VOCs accounted for about 60% of measured TOC. The missing species are suggested.

Total Organic Carbon is an interesting and worthwhile measurement target and I think the approach the authors have taken here is reasonable. The analysis of the data is clear and appropriate. The close value of the overall total OC in marine air to other values reported in the literature is an interesting result.

**Major questions**
The authors are attempting to measure a small value (~10ppb of "missing" carbon) on top of a large, imprecise background (400 ppm of atmospheric CO2 with a precision of 100 ppb). I think this is possible with a lot of time-averaging over a period with stable concentrations. However there is very little recognition and discussion of the difficulties associated with making a highly precise measurement atop a large background. For example, it is stated that methane is combusted with 98.7% efficiency; however, at typical atmospheric mixing ratios this corresponds to 40-50ppb which is much larger than the target VOC concentrations. CO was not measured, and the uncertainty in the reconstructed CO (~6-8ppb) is on the same order as the suggested missing VOC.

Additionally, the authors need to much more clearly state the time-averaging period of not just the new instrument presented here but also of the component instruments, and the previous TOC instruments cited from the literature. Otherwise it is not possible to assess and compare the various detection limits. For instance, the precision of the TOC measurement, which involves subtracting total atmospheric CO2 and CH4, can only be as precise as the precision of the direct CO2 and CH4 measurements. The CO2 measurement has a precision of 100 ppb at 2Hz. A best-case hourly precision of 8 ppb is stated, are the data presented hourly data? I understand the actual hourly precision was 30 ppb, is an 8ppb difference in speciated VOC compared to measured TOC significant with this precision? Does the calculation of the hourly precision take into account the instrument duty cycle (2 minutes ambient followed by 1 minute catalyst)?

**Specific comments**
32      - This is actually an estimate of the total number of species that have been measured. The extant number of species in the atmosphere is higher.

53      - the VOC relationship to ozone and organic aerosol (both climate forcers) is another important climate-related consideration.

127      The VOC concentrations in various analytical standards are exceedingly high compared to the range of VOC measured, and the suggested amount of "missing" carbon.
Gas standard: 1295 ppb; instrument measures 63ppbv lower
Background: 39 ppb
Typical total OC: 19 ppb
Why were such high values of calibration standard concentration chosen? I strongly suggest that the instrument is re-calibrated in a more appropriate range.

151      Can you please show more clearly how these two values were calculated (35ppb and 8ppb).

Fig 2      It would be useful to have a figure that shows the measured TOC compared to known TOC, for the multiple-step calibration with the 6-component calibration standard shown in Figure 2.

Figures  The paper is missing a figure showing a time series or diurnal cycle of total measured OC, minus CO, on the same scale and the same plot as total measured speciated VOC. As the difference between these two measurements is a major result of the paper, this plot needs to be shown.

---

## Short Comment (SC1) · 26 Nov 2018

Yang and Fleming are to be commended for tackling the challenge of measuring total organic carbon (TOC) in a manner that promises to bypass some of the limitations of previous attempts. Having thought about this for some decades, it seems clear that a catalytic conversion technique of this kind is the only way to capture the entire range of volatilities and functionalities inherent in the family of non-methane organic compounds. The authors correctly point out that high precision measurements are the key to being able to do this, since ambient $CO_2$, $CH_4$, and $CO$ need to be subtracted from a total signal to obtain TOC. After consideration of the details in the manuscript, I

would like to suggest various adjustments to the inlet design/material and catalyst material/configuration to potentially improve the measurement performance and precision. Based on my previous experience designing inlets for ambient sampling, the hot catalyst surface should be the first surface that the sample flow encounters, as semi- and low-volatility VOCs and particle-bound VOCs will reversibly partition to inlet surfaces and cause delayed instrument responses and effect quantification. The authors should consider mounting the catalyst system at the front of the analysis system to minimize these effects. Secondly, we have found, in our previous research (Veres et al., 2010), that the platinum catalysts of the type used by Yang and Fleming work well for small flows, however result in undesirable flow restrictions and pressure drops when used for applications that demand larger flows. Consequently, we developed a catalyst based on platinum screens that allowed much higher sample flows (Stockwell et al., 2018). Lastly, Teflon PFA is permeable to $CO_2$, which results in increased instrumental error due to a drifting and flow-path dependent backgrounds that are difficult to capture. As a result, we have converted parts of the system to nonpermeable stainless steel or Synflex 1300 (sometimes called Dekabon) tubing (no endorsement implied, there are probably others that would work). We offer these suggestions, based on both our published laboratory development of similar techniques as well as our unpublished results that were core to the design processes in our systems, in hopes that it may help to further improve the promising technique described in the work by Yang and Fleming. We look forward to seeing how this technique works with an on-line CO measurement added to the experiment.

References

Stockwell, C. E., Kupc, A., Witkowski, B., Talukdar, R. K., Liu, Y., Selimovic, V., Zarzana, K. J., Sekimoto, K., Warneke, C., Washenfelder, R. A., Yokelson, R. J., Middlebrook, A. M., and Roberts, J. M.: Characterization of a catalyst-based conversion technique to measure total particle nitrogen and organic carbon and comparison to a particle mass measurement instrument, Atmos. Meas. Tech., 11, 2749-2768, 2018.

Veres, P. R., Gilman, J. B., Roberts, J. M., Kuster, W. C., Warneke, C., Burling, I. R., and de Gouw, J.: Development and validation of a portable gas phase standard generation and calibration system for volatile organic compounds, Atmos. Chem. Phys., 3, 683-691., 2010.

---

## Author Comment (AC1) · 28 Nov 2018

Author Comment with regard to:

**"Estimation of atmospheric total organic carbon (TOC) – paving the path towards carbon budget closure"**

**by M. Yang and Z. Fleming**

Many thanks for the thoughtful *comments and suggestions from Anonymous Referee #1.* We are very glad to hear that the referee found our contribution valuable, our measurement approach interesting and scalable elsewhere. Below are our replies to the referee's comments, which are in *italic*.

*Anonymous Referee #1*
*The authors employ a new catalyst based approach to measuring TOC+CO in ambient air (with co-location of high-precision measurements of CO2 and CH4 which allow the subtraction of these species). Measurements of TOC are much-needed and though there are limitations to these specific measurements, this manuscript describes an interesting new approach to TOC measurements and a straight-forward application to a marine site.*

*I have only minor suggestions and requests for clarification:*
*1. Given the novel measurement approach it seems this should be described in the abstract (brief description, precision, comment on whether all species are comprehensively detected – see comment #2).*
Thanks for the suggestions. We will add the following sentences to the abstract (after the first sentence, line 13).
"Here we present a novel and simple approach to measure atmospheric non-methane total organic carbon (TOC) based on catalytic oxidation of organics in bulk air to carbon dioxide. This method shows little sensitivity towards humidity and near 100% oxidation efficiencies for all VOCs tested. We estimate a best-case hourly precision of 8 ppb C during times of low ambient variability in carbon dioxide, methane, and carbon monoxide (CO). As proof of concept of this approach, we show measurements of TOC+CO during August-September 2016 from a coastal city in the southwest United Kingdom."

We will add this sentence to the second to last line of the abstract:

"Finally, we note that the use of a short, heated sample tube can improve the transmission of organics to the analyzer, while operating our system alternately with and without a particle filter should enable a better separation of semi-volatile and particulate organics from the VOCs within the TOC budget."

*2. Section 2: The authors mention the inlet and briefly allude to SV/IVOCs and aerosols in the text (lines 100, 123-126). The manuscript would benefit from more discussion of this, but most importantly, the authors should reiterate these gaps in the conclusions and abstract. Ultimately the reported TOC is not comprehensive and this should be made clear to the reader, with appropriate suggestions for assessing the degree of*

*comprehensiveness and/or improving the instrumentation in the future (as given on lines 308-309).*
See above. Also, we will add the following sentence to line 308:
"Due to the use of a long, unheated Teflon inlet tube, the semi-volatile and particulate organics were likely not well represented in the measurements presented here. A simple improvement to this measurement system would be to minimize the length of the inlet tube before the catalyst."

*3. Section 2.1: There is some ambiguity of units in this section between ppb and ppbC and it would be useful for the units of all quantities to be carefully defined (line 115). I believe that some quantities are incorrectly given as ppb instead of ppbC (line 131, 135, 141, 145, 148, 151, 155), though it's not always clear from the text. Please correct as necessary.*
We apologize for the ambiguity. The units are:
Line 115: ppb C and ppb are equivalent here
Line 131, 135, 141, 145, 151, 155: ppb C
Line 148: ppb C and ppb are equivalent here
These units are specified in the revision.

*4. line 84: I suggest you place "e.g" in front of the Nolscher et al. reference since many studies have discussed the "missing OH reactivity"*
Suggestion accepted.

*5. line 194: September 8-10 also looks windy and rainy from the plots. Why aren't these dates included here?*
These dates will be added to this sentence in the revision.

*6. Figure captions: I recommend adding the measurement location to the captions so that the casual reader is clear that these represent field measurements at a given site.*
Suggestion accepted.

*7. Figure 7, Figure 9, and lines 225-226, 250, 263: These scatter plots show some relationships, but the correlation appears quite weak. Please include the R2 on the figures and temper the text accordingly.*
The correlations in the scatter plots are heavily influenced by the random noise in the measurement. Take Figure 9, for example, (TOC+CO vs. Sum(VOC)). The r2 value is 0.12 for the hourly average (N=299), 0.20 for the 6-hr average (N=61), and 0.91 for the bin average shown in Figure 9 (N=6). All of these correlations are highly significant at the 95% confidence level according to a t-test. The purpose of bin-averaging is to remove most of the random noise in order to more clearly demonstrate the mean relationship. We will add this information to the revision.

---

## Author Comment (AC3) · 28 Nov 2018

Thanks a lot for your encouraging words and valuable insights. We will include the references you've suggested in our revised manuscript.
* * *

---

## Author Response (AR1)

Dear editor:

Thank you for your efforts in coordinating the reviews for "Estimation of atmospheric total organic carbon (TOC) – paving the path towards carbon budget closure." Please find below our point-by-point replies to the referee's comments, which

5    are in *italic*.

***Anonymous Referee #1***

*1. Given the novel measurement approach it seems this should be described in the abstract (brief description, precision, comment on whether all species are comprehensively detected – see comment #2).*
10   Thanks for the suggestions. We will add the following sentences to the abstract (after the first sentence, line 13).
"Here we present a novel and simple approach to measure atmospheric non-methane total organic carbon (TOC) based on catalytic oxidation of organics in bulk air to carbon dioxide. This method shows little sensitivity towards humidity and near 100% oxidation efficiencies for all VOCs tested. We estimate a best-case hourly precision of 8 ppb C during times of low ambient variability in carbon dioxide, methane, and carbon monoxide (CO). As proof of concept of this approach, we show
15   measurements of TOC+CO during August-September 2016 from a coastal city in the southwest United Kingdom."

We have added this sentence to the second to last line of the abstract:

"Finally, we note that the use of a short, heated sample tube can improve the transmission of organics to the analyzer, while
20   operating our system alternately with and without a particle filter should enable a better separation of semi-volatile and particulate organics from the VOCs within the TOC budget."

*2. Section 2: The authors mention the inlet and briefly allude to SV/IVOCs and aerosols in the text (lines 100, 123-126). The manuscript would benefit from more discussion of this, but most importantly, the authors should reiterate these gaps in the*
25   *conclusions and abstract. Ultimately the reported TOC is not comprehensive and this should be made clear to the reader, with appropriate suggestions for assessing the degree of comprehensiveness and/or improving the instrumentation in the future (as given on lines 308-309).*
See above. Also, we have added the following sentence to line 308:
"Due to the use of a long, unheated Teflon inlet tube, the semi-volatile and particulate organics were likely not well represented
30   in the measurements presented here. A simple improvement to this measurement system would be to minimize the length of the inlet tube before the catalyst."

*3. Section 2.1: There is some ambiguity of units in this section between ppb and ppbC and it would be useful for the units of all quantities to be carefully defined (line 115). I believe that some quantities are incorrectly given as ppb instead of ppbC (line 131,*
35   *135, 141, 145, 148, 151, 155), though it's not always clear from the text. Please correct as necessary.*
We apologize for the ambiguity. The units are:
Line 115: ppb C and ppb are equivalent here
Line 131, 135, 141, 145, 151, 155: ppb C
Line 148: ppb C and ppb are equivalent here
40   These units are specified in the revision.

*4. line 84: I suggest you place "e.g" in front of the Nolscher et al. reference since many studies have discussed the "missing OH reactivity"*
Suggestion accepted.

45

*5. line 194: September 8-10 also looks windy and rainy from the plots. Why aren't these dates included here?*
These dates have been be added to this sentence in the revision.

*6. Figure captions: I recommend adding the measurement location to the captions so that the casual reader is clear that these*
50   *represent field measurements at a given site.*
Suggestion accepted.

*7. Figure 7, Figure 9, and lines 225-226, 250, 263: These scatter plots show some relationships, but the correlation appears quite weak. Please include the R2 on the figures and temper the text accordingly.*
55   The correlations in the scatter plots are heavily influenced by the random noise in the measurement. Take Figure 9, for example, (TOC+CO vs. Sum(VOC)). The r2 value is 0.12 for the hourly average (N=299), 0.20 for the 6-hr average (N=61), and 0.91 for the bin average shown in Figure 9 (N=6). All of these correlations are highly significant at the 95% confidence level according

to a t-test. The purpose of bin-averaging is to remove most of the random noise in order to more clearly demonstrate the mean relationship. We have added this information to the revision.

60

*Anonymous Referee #2*

*The authors are attempting to measure a small value (~10ppb of "missing" carbon) on top of a large, imprecise background (400 ppm of atmospheric CO2 with a precision of 100 ppb). I think this is possible with a lot of time-averaging over a period with stable concentrations. However there is very little recognition and discussion of the difficulties associated with making a highly precise measurement atop a large background. For example, it is stated that methane is combusted with 98.7% efficiency; however, at typical atmospheric mixing ratios this corresponds to 40-50ppb which is much larger than the target VOC concentrations. CO was not measured, and the uncertainty in the reconstructed CO (~6-8ppb) is on the same order as the suggested missing VOC.*

The large backgrounds in the major gases (especially CO2) are definitely the most important sources of random uncertainty in the TOC measurement. The very high precision/stability of the CO2/CH4 instrument and the frequent alternation between the ambient air and catalyst-scrubbed air measurement (enabling further averaging) help to largely overcome these challenges. We note that in equation 1:

TOC + CO = CO2* + CH4* - CO2 – CH4

The small amount (1.3%) of yet-to-be oxidized CH4 is accounted for in the CH4* term. Furthermore, any errors in the 'zeros' of the CO2 and CH4 measurements should be canceled out in the equation above.

That CO was not measured at the time of the TOC+CO/PTRMS measurements is a shortcoming of this dataset. Appropriately, we do not claim to have closed the organic carbon budget in the title, abstract or conclusion. Rather, we mostly focus on presenting a new way to quantify TOC.

80    *Additionally, the authors need to much more clearly state the time-averaging period of not just the new instrument presented here but also of the component instruments, and the previous TOC instruments cited from the literature. Otherwise it is not possible to assess and compare the various detection limits. For instance, the precision of the TOC measurement, which involves subtracting total atmospheric CO2 and CH4, can only be as precise as the precision of the direct CO2 and CH4 measurements. The CO2 measurement has a precision of 100 ppb at 2Hz. A best-case hourly precision of 8 ppb is stated, are the data presented*

85    *hourly data? I understand the actual hourly precision was 30 ppb, is an 8ppb difference in speciated VOC compared to measured TOC significant with this precision? Does the calculation of the hourly precision take into account the instrument duty cycle (2 minutes ambient followed by 1 minute catalyst)?*

Yes we have accounted for the instrument duty cycle in the hourly precision calculation. Specifically, consecutive ambient air

90    (last 5 seconds of 2 minutes) & catalyst (last 25 seconds out of 1 minute) are treated together as a measurement pair. Random uncertainty scales with sqrt(N), where N = number of measurements. Thus starting from a precision of 100 ppb at 2Hz for CO2, we get sqrt((100 ppb/sqrt(10))^2+(100 ppb/sqrt(50))^2)
 = 34.6 ppb for each pair of measurement. There are 20 measurement pairs in an hour, and so the hourly precision becomes 34.6 ppb/sqrt(20)= 7.7 ppb (see also line 151). The precision of CH4 measurements contributes negligibly to the uncertainty of the

95    TOC measurement.

However, the above calculation represents a best-case scenario because it assumes that the major gases (mostly CO2) are invariable. When these major gases are changing rapidly, the calculation of CO2* + CH4* - CO2 – CH4 will natural yield highly variable TOC + CO estimates. The 30 ppb mentioned on line 155 is not a precision; it's the actual standard deviation in a

100   subset of the measurements and an indication that the major gases at this semi-polluted coastal city were never especially constant. The comparison between the estimated TOC and Sum(VOC) is made in the mean for all Atlantic-dominated airmasses (142 hours total), and not on an hour-by-hour basis.

Finally, previous TOC measurements were based on GC analyses and have a cycle time of tens of minutes. We have added the

105   above information in the revised manuscript.

**Specific comments**
*32 - This is actually an estimate of the total number of species that have been measured. The extant number of species in the atmosphere is higher.*

110   We have revised the sentence to "The total number of measured organic carbon species…"

*53 - the VOC relationship to ozone and organic aerosol (both climate forcers) is another important climate-related consideration.*
Suggestion accepted.

115

*127 The VOC concentrations in various analytical standards are exceedingly high compared to the range of VOC measured, and the suggested amount of "missing" carbon.*
*Gas standard: 1295 ppb; instrument measures 63ppbv lower*
*Background: 39 ppb*
*Typical total OC: 19 ppb*
*Why were such high values of calibration standard concentration chosen? I strongly suggest that the instrument is re-calibrated in a more appropriate range.*

We note that there is an uncertainty in the diluted gas standard (up to 78 ppb), which mostly comes from the certified certainty of the standard itself (from Apel-Riemer Environmental). The measurement of TOC system agrees with the gas standard concentration within the uncertainties. We purposely used a high standard concentration because if all the VOCs are fully oxidized at concentrations 1-2 orders of magnitude higher than ambient, we would expect these VOCs in ambient air to fully oxidize too.

*151 Can you please show more clearly how these two values were calculated (35ppb and 8ppb).*
See detailed derivations above.

*Fig 2 It would be useful to have a figure that shows the measured TOC compared to known TOC, for the multiple-step calibration with the 6-component calibration standard shown in Figure 2.*
Perhaps this was not very clear in the text, but the measurements of the oxidized gas standard by the TOC system and by the PTRMS were done separately. To measure this oxidized standard with both systems at the same time would require an approximate doubling of the gas flow rate through the catalyst, which will probably reduce the oxidation efficiency.

*Figures The paper is missing a figure showing a time series or diurnal cycle of total measured OC, minus CO, on the same scale and the same plot as total measured speciated VOC. As the difference between these two measurements is a major result of the paper, this plot needs to be shown.*
We have produced a plot showing the diurnal cycles of the estimated TOC and Sum(VOC) on the same scale (see below). We limited this comparison to a) Atlantic dominated airmasses only (so that we could use the CO concentration measured from Cardiff), and b) for weekends only (to reduce the effect of location pollution). We see that in this subset of data (about 200 hrs total), Sum(VOC) shows essentially no diurnal variability. The TOC estimate is noisier, with a suggestion of higher concentrations in the early afternoon. The minima in TOC at around 0900 and 1700 UTC are most likely artifacts – due to residual traffic signals in the Cardiff CO data despite the airmass selection. Having an in situ measurement of CO and a longer overlap between the measurement systems would lead to a more insightful comparison. We don't think this figure provides a lot of additional insight but can include it in the paper or in a supplementary if the referee/editor deems it necessary.

[Figure]

Other changes in the revision include:
- a slight re-organization of section 2.1 (moving the discussions on particulate and semi-volatile organics to the end of the section)
- additions of references suggested by James Roberts

The revised manuscript is attached below.

**Estimation of atmospheric total organic carbon (TOC) – paving the path towards carbon budget closure**

160    Mingxi Yang[1*], Zoë L. Fleming[2&]

[1] Plymouth Marine Laboratory, Plymouth, United Kingdom

[2] National Centre for Atmospheric Science (NCAS), Department of Chemistry, University of Leicester, UK, United Kingdom

* Correspondence to M. Yang (miya@pml.ac.uk)

165    & Now at Center for Climate and Resilience Research (CR2), Departamento de Geofísica, Universidad de Chile, Santiago, Chile

**Abstract.** The atmosphere contains a rich variety of reactive organic compounds, including gaseous volatile organic carbon (VOCs), carbonaceous aerosols, and other organic compounds at varying volatility. Here we present a novel and simple

170    approach to measure atmospheric non-methane total organic carbon (TOC) based on catalytic oxidation of organics in bulk air to carbon dioxide. This method shows little sensitivity towards humidity and near 100% oxidation efficiencies for all VOCs tested. We estimate a best-case hourly precision of 8 ppb C during times of low ambient variability in carbon dioxide, methane, and carbon monoxide (CO). As proof of concept of this approach, we show measurements of TOC+CO during August-September

175    2016 from a coastal city in the southwest United Kingdom. TOC+CO was substantially elevated during the day on weekdays (occasionally over 2 ppm C) as a result of local anthropogenic activity. On weekends and holidays, with a mean (standard error) of 102 (8) ppb C, TOC+CO was lower and showed much less diurnal variability. TOC+CO was significantly lower when winds were coming off the Atlantic Ocean than when winds were coming off land if we exclude the weekday daytime. By subtracting the estimated CO from TOC+CO, we constrain the mean (uncertainty) TOC in marine air to be around 19 (±8) ppb C during

180    this period. A proton-transfer-reaction mass spectrometer (PTR-MS) was deployed at the same time, detecting a large range of organic compounds (oxygenated VOCs, biogenic VOCs, aromatics, dimethyl sulfide). The total speciated VOCs from the PTR-MS, denoted here as Sum(VOC), amounted to a mean (uncertainty) of 11(±3) ppb C in marine air. Possible contributions from a number of known organic compounds present in marine air that were not detected by the PTR-MS are assessed within the context of the TOC budget. Finally, we note that the use of a short, heated sample tube can improve the transmission of organics to the analyzer, while operating our system alternately with and without a particle filter should enable a better separation of semi-

[revised manuscript text omitted]
 (averaged from the last 25 seconds of the catalyst phase). Any errors in the 'zeros' of the $CO_2$ and $CH_4$ measurements should be canceled out in Equation 1. The monitoring of $CH_4$ enables the continuous assessment of the efficiency of the catalytic conversion (computed as $[CH_4 - CH_4*]/ CH_4$). At the flow and temperature used, oxidation of $CH_4$ was highly efficient (98.7-98.9%) and largely insensitive towards humidity during this campaign (Figure 1). $CH_4$ is thermodynamically one of the most reduced and stable compounds. Thus its rapid and near complete removal by the catalyst suggests ~100% oxidation of other VOCs as well as CO to $CO_2$, which is confirmed by laboratory tests as discussed below. We note that based on previous work using platinum catalytic converters (Veres et al. 2010; Stockwell et al. 2018), it is likely that the oxidation efficiency of this system will be lower at a much higher flow rate.

To verify the TOC system, we measured a diluted VOC gas mix. A multi-species gas standard consisting of methanol, acetaldehyde, acetone, DMS, benzene, and toluene (nominal mixing ratio of 500 ppb for each VOC balanced in nitrogen, Apel-Riemer Environmental, Inc, USA) was diluted by a factor of 10 with zero air. The zero air was generated by pre-scrubbing a low-VOC synthetic air (BOC BTCA 178, containing 20% oxygen) with a second 450°C platinum catalyst.

The expected total ppb of carbon in this diluted standard (1295 ± ≤78 ppb C) is computed as follows:

$$Sum(VOC) = \sum VOC \cdot N_c \qquad (2)$$

Here $N_c$ is the number of carbon in each speciated VOC. The total uncertainty in this Sum(VOC) is propagated from the accuracies of the VOC standard concentrations and from the uncertainties in the dilution. We measured a difference in TOC + CO between the diluted VOC standard and zero air alone of 1232 (± 1 standard error of 21) ppb C. Assuming negligible CO in the zero air as well as in the VOC standard, TOC + CO here can simply be equated to TOC. In this case TOC and Sum(VOC) agree well within the experimental uncertainties. Here we have purposely chosen a high VOC standard concentration because if all the VOCs are fully oxidized at concentrations 1 to 2 orders of magnitude higher than in ambient air, we would expect these VOCs at ambient levels to fully oxidize as well.

Because the catalyst (made up of platinum, glass wool, and stainless steel) does not contain any carbonaceous components, we expected the instrument background in TOC + CO (i.e. when measuring air that is free of organics and CO) to

Mingxi Yang 11/29/18 11:48 AM
Mingxi Yang 11/29/18 11:48 AM
Mingxi Yang 11/27/18 12:25 PM
Mingxi Yang 11/27/18 10:48 AM
Mingxi Yang 11/27/18 10:48 AM
Mingxi Yang 11/27/18 10:48 AM
Mingxi Yang 11/27/18 10:47 AM
Mingxi Yang 11/27/18 10:48 AM
Mingxi Yang 11/27/18 9:40 AM
Deleted: In addition to gases, ambient TOC measured with this method likely includes some aerosols and low/moderate-volatility compounds. The contribution of particulate and semi-volatile organics towards TOC depends on their transmission through the inlet tube as well as on their oxidation efficiency in the catalyst. We did not test these aspects as organic aerosol mass is already quantifiable using aerosol mass spectrometry as well as thermal methods (e.g. Sunset Laboratory's OCEC analyzer), and thus not the focus of this work.
Mingxi Yang 11/27/18 9:52 AM
Mingxi Yang 11/27/18 11:44 AM
Mingxi Yang 11/27/18 11:44 AM
Mingxi Yang 11/27/18 11:45 AM
Mingxi Yang 11/27/18 11:45 AM
Mingxi Yang 11/27/18 11:45 AM
Mingxi Yang 11/27/18 11:45 AM
Mingxi Yang 11/27/18 11:45 AM
Mingxi Yang 11/27/18 11:45 AM
Mingxi Yang 11/27/18 11:45 AM
Mingxi Yang 11/27/18 11:44 AM

be zero.  However, post-campaign measurements of zero air (see above) yielded a TOC + CO background of 39 (± 1 standard

305  error of 3) ppb C.  The reason for this small but significantly positive background is unclear.  It could be that some particulate

organic carbon either preexisting in the atmosphere or formed via charring to 450°C is captured by the glass wool, and then

slowly oxidized to $CO_2$ over time.  We note that the Sunset Laboratory's OCEC (Organic Carbon Elemental Carbon) analyzer

heats to 850°C (over manganese dioxide) for complete desorption and conversion of refractory organics (e.g. soot) to $CO_2$.  We

subtracted the background value of 39 ppb C from the TOC + CO measurements during the 2016 campaign.  However, the fact

310  that the background measurement was not made at the time of the campaign is a source of potential bias in this dataset.

Measurement of TOC+CO by our approach is made possible thanks to the very high precision of the Picarro G2311f

instrument (~100 and 0.4 ppb for $CO_2$ and $CH_4$ at 2 Hz, respectively).  Scatter in TOC+CO depends on random noise as well as

ambient variability in the $CO_2$/$CH_4$/CO mixing ratios, with $CO_2$ being the most critical.  The measurement precision is

significantly improved through averaging, as random noise scales with $N^{1/2}$ (N being the number of measurement).  In the limit

315  of no ambient variability in $CO_2$/$CH_4$/CO, each pair of $CO_2$*/ $CH_4$* and $CO_2$ /$CH_4$ measurements has a propagated precision of

35 ppb C (= $((100\ ppb/N^{1/2})^2 + (100\ ppb/N^{*1/2})^2)^{1/2}$) in our setup.  Since there are nominally 20 measurements per hour, this

implies a best-case hourly precision of 8 ppb C (=35 ppb C/$20^{1/2}$) for TOC + CO.  While not substantially more precise than

earlier methods, the technique described here is robust in that it avoids many of the uncertainties and complexities associated

with trapping and desorption (e.g. Roberts et al. 1998).  The precision estimate above may be appropriate for parts of the remote

320  marine atmosphere (i.e. very low variability in $CO_2$/$CH_4$/CO).  Closer to emission sources, the greater variability in these major

gases is expected to significantly increase the scatter in TOC + CO.  At our polluted coastal environment where the major gases

were generally not very constant, the standard deviation in the hourly mean TOC + CO was about 30 ppb C during periods of

fairly low ambient $CO_2$ variability (1 standard deviation of ~0.2 ppm).

In addition to gases, ambient TOC measured with this method likely includes some aerosols and low/moderate-volatility

325  compounds, which can reversibly partition onto the inlet tube surface and lengthen the measurement response time.  The

contribution of these particulate and semi-volatile organics towards TOC depends on their transmission through the inlet tube as

well as on their oxidation efficiency in the catalyst.  We did not explicitly test these aspects as organic aerosol mass is already

quantifiable using aerosol mass spectrometry as well as thermal methods (e.g. Sunset Laboratory's OCEC analyzer).

330  **2.2 Speciated VOC measurements**

Speciated organic gases were quantified using a PTR-MS, which was freshly serviced and calibrated by Ionicon.  The PTR-MS

settings were essentially the same as those used by Yang et al. (2013; 2014), except for a lower drift tube pressure (2.25 mbars).

The monitored masses (m/z) with a $H_3O^+$ source were attributed to the following compounds: m/z 33 (methanol), 42

Mingxi Yang 11/27/18 10:52 AM

Mingxi Yang 11/27/18 10:54 AM

Mingxi Yang 11/27/18 10:55 AM

Mingxi Yang 11/27/18 10:55 AM

Mingxi Yang 11/27/18 10:54 AM

Mingxi Yang 11/27/18 11:03 AM

Mingxi Yang 11/27/18 12:07 PM

Mingxi Yang 11/27/18 10:56 AM

Mingxi Yang 11/27/18 10:57 AM

Mingxi Yang 11/27/18 10:58 AM

Mingxi Yang 11/27/18 10:56 AM

Mingxi Yang 11/27/18 10:56 AM

Mingxi Yang 11/27/18 10:56 AM

Mingxi Yang 11/27/18 11:00 AM

Mingxi Yang 11/27/18 11:33 AM

Mingxi Yang 11/27/18 11:09 AM

Mingxi Yang 11/27/18 11:34 AM

Mingxi Yang 11/27/18 11:09 AM

Mingxi Yang 11/27/18 11:24 AM

Mingxi Yang 11/27/18 11:09 AM

(acetonitrile), 43 (fragmented propanol or acetic acid ), 45 (acetaldehyde), 47 (ethanol), 59 (acetone), 61 (propanol or acetic acid

[revised manuscript text omitted]
. Similar to Figure 7, the degree of correlation here is heavily influenced by the random noise in the TOC+CO measurement. The $r^2$ value between TOC+CO and Sum(VOC) is 0.12 for the hourly average (N=299), 0.20 for the 6-hr average (N=61), and 0.91 for the bin average as shown in Figure 9 (N=6). All of these correlations are highly significant at the 95% confidence level according to a t-test. Bin-averaging helps to remove most of the random noise in order to more clearly demonstrate the mean relationship.

We see that TOC (19±≥8 ppb C, estimated by subtracting CO from TOC+CO) is ~70% higher than Sum(VOC) when winds were from the southwest but the difference is within the measurement uncertainties. Comparing TOC with Sum(VOC) in background marine air, partly necessitated here by our lack of in situ CO observations, challenges the signal to noise of the TOC measurement. We expect such a comparison to be more insightful in environments with a higher TOC burden. Nevertheless, we know that the PTR-MS with hydronium ion source is not suitable for detecting many VOCs, such as low molecular weight hydrocarbons that are expected to make up most of the primary anthropogenic organic emissions. Below we examine the magnitudes of some nominally abundant VOCs in marine air that were not measured by the PTR-MS.

Formaldehyde (HCHO), the most abundant aldehyde in the atmosphere, was measured at PML's PPAO from the spring of 2015 to the beginning of 2016 using multi-axes differential optical absorption spectroscopy (MAX-DOAS). During southwesterly conditions the surface mixing ratio of HCHO was about 0.5 ppb C (Johannes Lampel, personal communication in 2016). Non-methane hydrocarbons, such as alkanes, have large seasonal variability in temperate regions, with significantly lower abundance in the summer time due to greater OH destruction. Grant et al. (2011) reported long-term time series measurements of hydrocarbons from Mace Head. For the months of August and September from 2005 to 2009, the mean mixing ratio of ethane and propane were about 2 and <0.3 ppb C during maritime conditions, respectively. Grant et al. (2011) also measured (i- and n-) butane and (i- and n-) pentane, which were of even lower abundance. Salisbury et al. (2001) reported the mixing ratios of a large range of alkenes from Mace Head. The most abundant alkenes, ethene and propene, had mean mixing ratios of about 0.05 and 0.06 ppb C, respectively. Acetylene, the simplest alkyne, has a mixing ratio of ~0.5 ppb C over the Atlantic (Xiao et al., 2007). The ocean is a source of a host of halocarbons. Among these chloromethane ($CH_3Cl$) is the most abundant, with a mixing ratio of the order of 0.5 ppb C in the marine boundary layer (Yokouchi et al., 2013). Together these organic gases make up to approximately 5 ppb C.

Because of the high temperature and oxidative conditions in the catalyst, we expect some organic aerosols and semi-volatile species to be oxidized and detected as $CO_2$. Particulate matter less than 2.5 μm (PM2.5) in the City Center of Plymouth, also from the Defra Air Quality Monitoring station, was about 6 μg m$^{-3}$ during this period when winds were from the southwest. If we assume that half of the PM2.5 was made up of carbon by mass (most likely an overestimate for this region, e.g. Morgan et al., 2010), the aerosol contribution to TOC could be up to 6 ppb C (or a third of TOC). Heald et al. (2008) reported that organic aerosols only accounted for 4-16% of total speciated organics (gases plus aerosols) in the marine atmosphere. Overall, considering the aforementioned VOCs and organic aerosols that were not detected by the PTR-MS, there does not appear to be a substantial 'missing' term in the TOC mass budget.

**6 Concluding remarks**

In this paper we report a relatively novel and simple method to measure the mixing ratios of total organic carbon (TOC) and carbon monoxide in the atmosphere at a high frequency.  Based on essentially complete oxidation of organics in bulk air to $CO_2$ in a platinum catalyst, our method shows very low sensitivity towards ambient humidity, avoids the complexities associated with trapping and desorption, and has an hourly precision of as low as 8 ppb C.  Due to the use of a long, unheated Teflon inlet tube, the semi-volatile and particulate organics were likely not well represented in the measurements presented here.  A simple improvement to this measurement system would be to minimize the length of the inlet tube before the catalyst.  
[revised manuscript text omitted]

Stockwell, C. E., Kupc, A., Witkowski, B., Talukdar, R. K., Liu, Y., Selimovic, V., Zarzana, K. J., Sekimoto, K., Warneke, C., Washenfelder, R. A., Yokelson, R. J., Middlebrook, A. M., and Roberts, J. M.: Characterization of a catalyst-based conversion technique to measure total particle nitrogen and organic carbon and comparison to a particle mass measurement instrument, Atmos. Meas. Tech., 11, 2749-2768, 2018.

Tani A., Hayward, S., Hansel, A., Hewitt, C. N.: Effect of water vapour pressure on monoterpene measurements using proton transfer reaction- mass spectrometry (PTR-MS). Int J Mass Spectrom 239:161–169, 2004.

Veres, P. R., Gilman, J. B., Roberts, J. M., Kuster, W. C., Warneke, C., Burling, I. R., and de Gouw, J.: Development and validation of a portable gas phase standard generation and calibration system for volatile organic compounds, Atmos. Chem. Phys., 3, 683- 691., 2010.

Wang, Y., Munger, J., Xu, S., McElroy, M., Hao, J., Nielsen, C., and Ma, H.: $CO_2$ and its correlation with CO at a rural site near Beijing: implications for combustion efficiency in China, *Atmos. Chem. Phys*., 10, 8881-8897, https://doi.org/10.5194/acp-10-8881-2010, 2010.

Xiao, Y., Jacob, D. J., and Turquety S.: Atmospheric acetylene and its relationship with CO as an indicator of airmass age, *J. Geophys. Res*., 112, D12305, doi:10.1029/2006JD008268, 2007.

Yang, M., Beale, R., Smyth, T., and Blomquist, B.: Measurements of OVOC fluxes by eddy covariance using a proton-transfer-reaction mass spectrometer – method development at a coastal site, *Atmos. Chem. Phys.*, 13, 6165-6184, https://doi.org/10.5194/acp-13-6165-2013, 2013.

Yang, M., Beale, R., Liss, P., Johnson, M., Blomquist, B., and Nightingale, P.: Air–sea fluxes of oxygenated volatile organic compounds across the Atlantic Ocean, *Atmos. Chem. Phys.*, 14, 7499-7517, https://doi.org/10.5194/acp-14-7499-2014, 2014.

Yang, M., Bell, T., Hopkins, F., and Smyth, T.: Attribution of atmospheric sulfur dioxide over the English Channel to dimethyl sulfide and changing ship emissions, *Atmos. Chem. Phys.*, 16, 4771-4783, https://doi.org/10.5194/acp-16-4771-2016, 2016.

Yokouchi, Y., Inoue, J., and Toom-Sauntry D.: Distribution of natural halocarbons in marine boundary air over the Arctic Ocean, *Geophys. Res. Lett.*, 40, 4086–4091, *doi:*10.1002/grl.50734, 2013.

Zhao, J. and Zhang, R. Y.: Proton transfer reaction rate constants between hydronium ion (H3O(+)) and volatile organic compounds, Atmos. Environ., 38, 2177–2185, 2004.

[Figure]

Figure 1. Oxidation efficiency of $CH_4$ by the platinum catalyst was nearly 99% and demonstrated only a weak dependence on ambient humidity during the 1.5-month measurement campaign from the rooftop of Plymouth Marine Laboratory.

[Figure]

Figure 2. Catalytic oxidations of VOCs were complete, immediate, and independent of the VOC input within the range tested.

[Figure]

Figure 3. (A) Time series of TOC+CO in units of ppm C from the rooftop of Plymouth Marine Laboratory. Error bars on 6-hr means indicate standard errors; (B) wind speed (color-coded by wind direction) and rain rate.

565

[Figure]

Figure 4. (A) Averaged diurnal cycle in TOC+CO, and (B) $CO_2$ for weekdays, weekends/holidays, and all data during the 1.5-month measurement campaign from the rooftop of Plymouth Marine Laboratory. Error bars indicate standard errors. TOC+CO were much higher during weekdays than weekends, especially in the daytime. Limited diel variability in TOC+CO was observed in the weekend data. $CO_2$ was also higher in the daytime during weekdays than weekends. UTC here is one hour behind local time.

570

Mingxi Yang 11/27/18 10:01 AM

Mingxi Yang 11/27/18 10:01 AM

[Figure]

Figure 5. Weekday-weekend difference in TOC+CO (ppm C) vs. weekday-weekend difference in $CO_2$ from the rooftop of Plymouth Marine Laboratory. A positive correlation is observed, with a dimensionless ratio ranging from mostly less than 0.05:1 to over 0.2:1.

575

[Figure]

Figure 6. (A) TOC+CO and $CO_2$, and (B) Sum(VOC) and speciated VOC averaged to 10-deg wind direction bins. Error bars indicate standard errors. TOC+CO, $CO_2$, and Sum(VOC) showed higher values when winds were from land (north to southeast) than winds were from the ocean (southwest). Most VOCs had lower mixing ratio in marine air than in air from land except for DMS. Amongst speciated VOCs measured by the PTR-MS, OVOCs dominated in terms of carbon mass. The fraction of carbon in the total speciated VOC mass was also lower in marine air.

580

[Figure]

585     Figure 7. Relationship between TOC+CO with (A) $CO_2$; (B) $O_3$; (C) percentage of time that the airmass was over the Atlantic ocean over the last 5 days; and (D) percentage of time that the airmass was over Mainland Europe and the English Channel over the last 5 days. Hourly data limited to weekends & weekday nights only.

590

[Figure]

Figure 8. Time series of TOC + CO and Sum(VOC) from the rooftop of Plymouth Marine Laboratory. Sum(VOC) is shown as hourly mean and 6-hour median, while for clarity TOC + CO is shown as 6-hour median only.

[Figure]

Figure 9. TOC+CO correlated positively with Sum(VOC). Hourly data limited to weekends & weekday nights only.

---

## Author Response (AR2)

Dear editor:

Thank you for your suggestion. Please find below our point-by-point replies to your comments, which are in *italic*.

*I am pleased to accept the revised manuscript for publication subject to a minor revision. The figure requested by Referee #2 showing a time series of TOC and sum(VOC) should be included in the manuscript. The error bar on TOC should be explained. Error bars on sum(VOC) should be added.*

We have added the figure showing the averaged diurnal cycles of Sum(VOC) and TOC (now Figure 10 in the manuscript) and
corresponding text in section 5.

**Estimation of atmospheric total organic carbon (TOC) – paving the**

**path towards carbon budget closure**

Mingxi Yang[1*], Zoë L. Fleming[2&]

[1] Plymouth Marine Laboratory, Plymouth, United Kingdom

[2] National Centre for Atmospheric Science (NCAS), Department of Chemistry, University of Leicester, UK, United Kingdom

* Correspondence to M. Yang (miya@pml.ac.uk)

& Now at Center for Climate and Resilience Research (CR2), Departamento de Geofísica, Universidad de Chile, Santiago, Chile

**Abstract.** The atmosphere contains a rich variety of reactive organic compounds, including gaseous volatile organic carbon (VOCs), carbonaceous aerosols, and other organic compounds at varying volatility. Here we present a novel and simple approach to measure atmospheric non-methane total organic carbon (TOC) based on catalytic oxidation of organics in bulk air to carbon dioxide. This method shows little sensitivity towards humidity and near 100% oxidation efficiencies for all VOCs tested. We estimate a best-case hourly precision of 8 ppb C during times of low ambient variability in carbon dioxide, methane, and carbon monoxide (CO). As proof of concept of this approach, we show measurements of TOC+CO during August-September 2016 from a coastal city in the southwest United Kingdom. TOC+CO was substantially elevated during the day on weekdays (occasionally over 2 ppm C) as a result of local anthropogenic activity. On weekends and holidays, with a mean (standard error) of 102 (8) ppb C, TOC+CO was lower and showed much less diurnal variability. TOC+CO was significantly lower when winds were coming off the Atlantic Ocean than when winds were coming off land if we exclude the weekday daytime. By subtracting the estimated CO from TOC+CO, we constrain the mean (uncertainty) TOC in Atlantic dominated airmasses to be around 23 (±≥8) ppb C during this period. A proton-transfer-reaction mass spectrometer (PTR-MS) was deployed at the same time, detecting a large range of organic compounds (oxygenated VOCs, biogenic VOCs, aromatics, dimethyl sulfide). The total speciated VOCs from the PTR-MS, denoted here as Sum(VOC), amounted to a mean (uncertainty) of 12(±≤3) ppb C in marine air. Possible contributions from a number of known organic compounds present in marine air that were not detected by the PTR-MS are assessed within the context of the TOC budget. Finally, we note that the use of a short, heated sample tube can improve the transmission of organics to the analyzer, while operating our system alternately with and without a particle filter should enable a better separation of semi-volatile and particulate organics from the VOCs within the TOC budget. 
[revised manuscript text omitted]
 (averaged from the last 25 seconds of the catalyst phase). Any errors in the 'zeros' of the $CO_2$ and $CH_4$ measurements should be canceled out in Equation 1. The monitoring of $CH_4$ enables the continuous assessment of the efficiency of the catalytic conversion (computed as $[CH_4 - CH_4*]/ CH_4$). At the flow and temperature used, oxidation of $CH_4$ was highly efficient (98.7-98.9%) and largely insensitive towards humidity during this campaign (Figure 1). $CH_4$ is thermodynamically one of the most reduced and stable compounds. Thus its rapid and near complete removal by the catalyst suggests ~100% oxidation of other VOCs as well as CO to $CO_2$, which is confirmed by laboratory tests as discussed below. We note that based on previous work using platinum catalytic converters (Veres et al. 2010; Stockwell et al. 2018), it is likely that the oxidation efficiency of this system will be lower at a much higher flow rate.

To verify the TOC system, we measured a diluted VOC gas mix. A multi-species gas standard consisting of methanol, acetaldehyde, acetone, DMS, benzene, and toluene (nominal mixing ratio of 500 ppb for each VOC balanced in nitrogen, Apel-Riemer Environmental, Inc, USA) was diluted by a factor of 10 with zero air. The zero air was generated by pre-scrubbing a low-VOC synthetic air (BOC BTCA 178, containing 20% oxygen) with a second 450°C platinum catalyst.

The expected total ppb of carbon in this diluted standard (1295 ± ≤78 ppb C) is computed as follows:

$$Sum(VOC) = \sum VOC \cdot N_c \tag{2}$$

Here $N_c$ is the number of carbon in each speciated VOC. The total uncertainty in this Sum(VOC) is propagated from the accuracies of the VOC standard concentrations and from the uncertainties in the dilution. We measured a difference in TOC + CO between the diluted VOC standard and zero air alone of 1232 (± 1 standard error of 21) ppb C. Assuming negligible CO in the zero air as well as in the VOC standard, TOC + CO here can simply be equated to TOC. In this case TOC and Sum(VOC) agree well within the experimental uncertainties. Here we have purposely chosen a high VOC standard concentration because if all the VOCs are fully oxidized at concentrations 1 to 2 orders of magnitude higher than in ambient air, we would expect these VOCs at ambient levels to fully oxidize as well.

Because the catalyst (made up of platinum, glass wool, and stainless steel) does not contain any carbonaceous components, we expected the instrument background in TOC + CO (i.e. when measuring air that is free of organics and CO) to be zero.  However, post-campaign measurements of zero air (see above) yielded a TOC + CO background of 39 (± 1 standard error of 3) ppb C.  The reason for this small but significantly positive background is unclear.  It could be that some particulate organic carbon either preexisting in the atmosphere or formed via charring to 450°C is captured by the glass wool, and then slowly oxidized to $CO_2$ over time.  We note that the Sunset Laboratory's OCEC (Organic Carbon Elemental Carbon) analyzer heats to 850°C (over manganese dioxide) for complete desorption and conversion of refractory organics (e.g. soot) to $CO_2$.  We subtracted the background value of 39 ppb C from the TOC + CO measurements during the 2016 campaign.  However, the fact that the background measurement was not made at the time of the campaign is a source of potential bias in this dataset.

Measurement of TOC+CO by our approach is made possible thanks to the very high precision of the Picarro G2311f instrument (~100 and 0.4 ppb for $CO_2$ and $CH_4$ at 2 Hz, respectively).  Scatter in TOC+CO depends on random noise as well as ambient variability in the $CO_2$/$CH_4$/CO mixing ratios, with $CO_2$ being the most critical.  The measurement precision is significantly improved through averaging, as random noise scales with $N^{1/2}$ (N being the number of measurement).  In the limit of no ambient variability in $CO_2$/$CH_4$/CO, each pair of $CO_2$*/ $CH_4$* and $CO_2$ /$CH_4$ measurements has a propagated precision of 35 ppb C (= $((100\ ppb/N^{1/2})^2+(100\ ppb/N^{*1/2})^2)^{1/2}$) in our setup.  Since there are nominally 20 measurements per hour, this implies a best-case hourly precision of 8 ppb C (=35 ppb C/$20^{1/2}$) for TOC + CO.  While not substantially more precise than earlier methods, the technique described here is robust in that it avoids many of the uncertainties and complexities associated with trapping and desorption (e.g. Roberts et al. 1998).  The precision estimate above may be appropriate for parts of the remote marine atmosphere (i.e. very low variability in $CO_2$/$CH_4$/CO).  Closer to emission sources, the greater variability in these major gases is expected to significantly increase the scatter in TOC + CO.  At our polluted coastal environment where the major gases were generally not very constant, the standard deviation in the hourly mean TOC + CO was about 30 ppb C during periods of fairly low ambient $CO_2$ variability (1 standard deviation of ~0.2 ppm).

In addition to gases, ambient TOC measured with this method likely includes some aerosols and low/moderate-volatility compounds, which can reversibly partition onto the inlet tube surface and lengthen the measurement response time.  The contribution of these 
[revised manuscript text omitted]

Mingxi Yang 12/15/18 9:39 AM
Mingxi Yang 12/15/18 9:39 AM

(65, 8) ppb C, while CO had a mean (median, SE) of 58 (51, 2) ppb. This implies that TOC in Atlantic-dominated airmasses averaged 23 ppb C, similar to the statistics selected by the southwest wind sector. In contrast, in airmasses dominated by mainland Europe and the English Channel (>50% relative residence time), TOC+CO had a mean (median) of 198 (191) ppb C (Figure 7D). Examples of these two types of airmasses as well as the regional map used for the airmass classification are shown in Figures S1-S3.

**5 Attempting a TOC budget closure**

During the 1.5 months study, the PTR-MS was used to measure a large range of organic gases over 22 days. The total mixing ratio of speciated organic carbon, Sum(VOC), is shown in Figure 8, which averaged 15 ppb C. Sum(VOC) was higher during weekdays (mean ± standard error of 16.0 ± 0.4 ppb C) than on weekends (mean ± standard error of 14.3 ±0.4 ppb C), but this difference is much less drastic than in the case of TOC+CO. This suggests that the significantly elevated TOC + CO during the weekday daytime was largely due to compounds that were not measured by the PTR-MS (e.g. CO, small alkanes, or alkenes).

Sum(VOC) was dominated by OVOCs (here methanol, acetone, acetaldehyde, ethanol, propanol/acid acid), consistent with Lewis et al. (2005) and Heald et al. (2008) (Figure 6B). Aromatic compounds (benzene, toluene, xylenes) were more abundant when winds were from land (northwest to northeast), as expected from anthropogenic emissions. Similarly, mixing ratios of biogenic VOCs (isoprene and monoterpenes) were higher when winds were from northwest to northeast in comparison to south and southwest. Among detected VOCs, only DMS mixing ratio was higher in marine air than in continental air. Similar to TOC+CO, Sum(VOC) showed the lowest value when winds were from the sea (~12 ppb C during southwesterly conditions). In comparison, in the synthesis by Heald et al. (2008) the sum of speciated organic compounds was about 8, 14, and 18 ppb C at Trinidad Head (California), Azores, and Chebogue Point (2004 measurements), respectively. Our mean Sum(VOC) is within the range of those coastal observations, which were generally more comprehensive than just the PTR-MS measurements here. Interestingly, the approximate carbon fraction of total VOCs (i.e. carbon mass : total mass) was also the lowest when winds were from the sea. This indicates that VOCs in marine air on average contain more non-carbon functional groups (e.g. nitrogen, sulfur).

Sum(VOC) correlated positively with TOC+CO in the mean (Figure 9) when limiting data to weekends and weekday nights only. Similar to Figure 7, the degree of correlation here is heavily influenced by the random noise in the TOC+CO measurement. The $r^2$ value between TOC+CO and Sum(VOC) is 0.12 for the hourly average (N=299), 0.20 for the 6-hr average (N=61), and 0.91 for the bin average as shown in Figure 9 (N=6). All of these correlations are highly significant at the 95% confidence level according to a t-test. Bin-averaging helps to remove most of the random noise in order to more clearly demonstrate the mean relationship.

Mingxi Yang 12/15/18 9:41 AM
Mingxi Yang 12/15/18 9:41 AM
Mingxi Yang 12/15/18 9:43 AM
Mingxi Yang 12/15/18 10:05 AM
Mingxi Yang 12/15/18 9:43 AM

Mingxi Yang 12/15/18 9:56 AM

Mingxi Yang 12/15/18 9:47 AM

Mingxi Yang 12/15/18 9:01 AM

The mean diurnal cycles of the estimated TOC and Sum(VOC) in Atlantic dominated airmasses and during weekends/holidays are shown in Figure 10. In this subset of data (131 hours total), Sum(VOC) shows essentially no diurnal variability. The TOC estimate is much noisier, with a suggestion of higher mixing ratios in the early afternoon. The minima in TOC at around 09:00 and 17:00 UTC are possibly artifacts – due to residual traffic signals in the Cardiff CO data despite the Atlantic airmass selection. While the average TOC is nearly twice the Sum(VOC) for Atlantic dominated airmasses and during weekends/holidays, TOC and Sum(VOC) differ by less than two times the SE of the TOC signal, implying that the two measurements are not significantly different. Comparing TOC with Sum(VOC) in the clean background marine air, partly necessitated here by our lack of in situ CO observations, challenges the signal to noise of the TOC measurement. We expect such a comparison to be more insightful in environments with a higher TOC burden. Nevertheless, we know that the PTR-MS with hydronium ion source is not suitable for detecting many VOCs, such as low molecular weight hydrocarbons that are expected to make up most of the primary anthropogenic organic emissions. Below we examine the magnitudes of some nominally abundant VOCs in marine air that were not measured by the PTR-MS.

      Formaldehyde (HCHO), the most abundant aldehyde in the atmosphere, was measured at PML's PPAO from the spring of 2015 to the beginning of 2016 using multi-axes differential optical absorption spectroscopy (MAX-DOAS). During southwesterly conditions the surface mixing ratio of HCHO was about 0.5 ppb C (Johannes Lampel, personal communication in

2016). Non-methane hydrocarbons, such as alkanes, have large seasonal variability in temperate regions, with significantly lower abundance in the summer time due to greater OH destruction. Grant et al. (2011) reported long-term time series measurements of hydrocarbons from Mace Head. For the months of August and September from 2005 to 2009, the mean mixing ratio of ethane and propane were about 2 and <0.3 ppb C during maritime conditions, respectively. Grant et al. (2011) also measured (i- and n-) butane and (i- and n-) pentane, which were of even lower abundance. Salisbury et al. (2001) reported the mixing ratios of a large range of alkenes from Mace Head. The most abundant alkenes, ethene and propene, had mean mixing ratios of about 0.05 and 0.06 ppb C, respectively. Acetylene, the simplest alkyne, has a mixing ratio of ~0.5 ppb C over the Atlantic (Xiao et al., 2007). The ocean is a source of a host of halocarbons. Among these chloromethane ($CH_3Cl$) is the most abundant, with a mixing ratio of the order of 0.5 ppb C in the marine boundary layer (Yokouchi et al., 2013). Together these organic gases make up to approximately 5 ppb C.

Because of the high temperature and oxidative conditions in the catalyst, we expect some organic aerosols and semi-volatile species to be oxidized and detected as $CO_2$. Particulate matter less than 2.5 μm (PM2.5) in the City Center of Plymouth, also from the Defra Air Quality Monitoring station, was about 6 μg m$^{-3}$ during this period when winds were from the southwest. If we assume that half of the PM2.5 was made up of carbon by mass (most likely an overestimate for this region, e.g. Morgan et al., 2010), the aerosol contribution to TOC could be up to 6 ppb C (or a third of TOC). Heald et al. (2008) reported that organic aerosols only accounted for 4-16% of total speciated organics (gases plus aerosols) in the marine atmosphere. Overall, considering the aforementioned VOCs and organic aerosols that were not detected by the PTR-MS, there does not appear to be a substantial 'missing' term in the TOC mass budget.

**6 Concluding remarks**

In this paper we report a relatively novel and simple method to measure the mixing ratios of total organic carbon (TOC) and carbon monoxide in the atmosphere at a high frequency. Based on essentially complete oxidation of organics in bulk air to $CO_2$ in a platinum catalyst, our method shows very low sensitivity towards ambient humidity, avoids the complexities associated with trapping and desorption, and has an hourly precision of as low as 8 ppb C. Due to the use of a long, unheated Teflon inlet tube, the semi-volatile and particulate organics were likely not well represented in the measurements presented here. A simple improvement to this measurement system would be to minimize the length of the inlet tube before the catalyst. Future measurements with and without an aerosol filter and a heated inlet should enable the semi-volatile and particulate fractions of TOC to be better separated from the VOCs.

The estimated TOC from a polluted marine environment is compared to the sum of speciated VOCs here. Accounting for literature values of unmeasured VOCs and organic aerosols, there does not appear to be a significant undetected fraction of organics in marine air. A more rigorous examination of the atmospheric organic carbon closure requires concurrent measurements of TOC, CO, and a comprehensive range of speciated organic compounds. Additional measurements of total OH reactivity would bridge the gap between organic burden and composition with oxidative capacity. With recent advances in mass spectrometry that are able to resolve ever more organic species (Hunter et al., 2017) as well as in chemical transport modeling (Safieddine et al., 2017), the stage seems set for closing the atmospheric organic budget.

**Author Contribution**

MY designed the TOC measurement system, carried out the field observations, and performed the data analysis. ZF led the airmass trajectory and dispersion modeling as well as the interpretation of those modeling results.

[revised manuscript text omitted]

Figure 6. (A) TOC+CO and $CO_2$, and (B) Sum(VOC) and speciated VOC averaged to 10-deg wind direction bins. Error bars indicate standard errors. TOC+CO, $CO_2$, and Sum(VOC) showed higher values when winds were from land (north to southeast) than winds were from the ocean (southwest). Most VOCs had lower mixing ratio in marine air than in air from land except for DMS. Amongst speciated VOCs measured by the PTR-MS, OVOCs dominated in terms of carbon mass. The fraction of carbon in the total speciated VOC mass was also lower in marine air.

[Figure]

Figure 7. Relationship between TOC+CO with (A) $CO_2$; (B) $O_3$; (C) percentage of time that the airmass was over the Atlantic ocean over the last 5 days; and (D) percentage of time that the airmass was over Mainland Europe and the English Channel over the last 5 days. Hourly data limited to weekends & weekday nights only.

[Figure]

Figure 8. Time series of TOC + CO and Sum(VOC) from the rooftop of Plymouth Marine Laboratory. Sum(VOC) is shown as hourly mean and 6-hour median, while for clarity TOC + CO is shown as 6-hour median only.

[Figure]

Figure 9. TOC+CO correlated positively with Sum(VOC). Hourly data limited to weekends & weekday nights only.

[Figure]

Figure 10. Averaged diurnal cycles in Sum(VOC) and estimated TOC in Atlantic dominated airmasses and during weekends/holidays.  Error bars indicate standard error.